# Switching state-space modeling of neural signal dynamics

**Mingjian He** [1,2◉], **Proloy Das** [1,3,4◉], **Gladia Hotan** [5], **Patrick L. Purdon** [1,3,4]*

**1** Department of Anesthesia, Critical Care and Pain Medicine, Massachusetts General Hospital, Boston, Massachusetts, United States of America, **2** Harvard-MIT Health Sciences and Technology, Massachusetts Institute of Technology, Cambridge, Massachusetts, United States of America, **3** Harvard Medical School, Boston, Massachusetts, United States of America, **4** Department of Anesthesia, Perioperative and Pain Medicine, Stanford University School of Medicine, Stanford, California, United States of America, **5** Institute of High Performance Computing, Agency for Science, Technology and Research (A*STAR), Singapore

◉ These authors contributed equally to this work.
* ppurdon@stanford.edu

## Abstract

Linear parametric state-space models are a ubiquitous tool for analyzing neural time series data, providing a way to characterize the underlying brain dynamics with much greater statistical efficiency than non-parametric data analysis approaches. However, neural time series data are frequently time-varying, exhibiting rapid changes in dynamics, with transient activity that is often the key feature of interest in the data. Stationary methods can be adapted to time-varying scenarios by employing fixed-duration windows under an assumption of quasi-stationarity. But time-varying dynamics can be explicitly modeled by switching state-space models, i.e., by using a pool of state-space models with different dynamics selected by a probabilistic switching process. Unfortunately, exact solutions for state inference and parameter learning with switching state-space models are intractable. Here we revisit a switching state-space model inference approach first proposed by Ghahramani and Hinton. We provide explicit derivations for solving the inference problem iteratively after applying a variational approximation on the joint posterior of the hidden states and the switching process. We introduce a novel initialization procedure using an efficient leave-one-out strategy to compare among candidate models, which significantly improves performance compared to the existing method that relies on deterministic annealing. We then utilize this state inference solution within a generalized expectation-maximization algorithm to estimate model parameters of the switching process and the linear state-space models with dynamics potentially shared among candidate models. We perform extensive simulations under different settings to benchmark performance against existing switching inference methods and further validate the robustness of our switching inference solution outside the generative switching model class. Finally, we demonstrate the utility of our method for sleep spindle detection in real recordings, showing how switching state-space models can be used to detect and extract transient spindles from human sleep electroencephalograms in an unsupervised manner.

**Data Availability Statement:** The authors confirm that all data underlying the findings are fully available without restriction. An implementation of the state inference and parameter learning algorithms described in this paper is available as part of SOMATA Python library on GitHub at

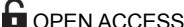

https://github.com/mh105/somata. SOMATA stands for State-space Oscillator Modeling And Time-series Analysis. All code written in support of this publication, simulation scripts, and analyzed sleep EEG recordings is available on Zenodo at https://doi.org/10.5281/zenodo.7818644.

**Funding:** This work is funded by National Institutes of Health (NIH) grant R01AG054081; received by PLP. The funders had no role in study design, data collection and analysis, decision to publish, or preparation of the manuscript.

**Competing interests:** The authors have declared that no competing interests exist.

## Author summary

An inherent aspect of brain activity is that it changes over time, but existing methods for analyzing neuroscience data typically assume that the underlying activity is strictly stationary, i.e., the properties of that activity do not change over time. One way of handling time-varying data is to break the data into smaller segments that one assumes to be quasi-stationary, but this approach only works if signals vary gradually, and tends to perform poorly when changes are rapid or the target activity is transient in nature. A class of models called linear switching state-space models can explicitly represent time-varying activity, but they pose another set of challenges: exact solutions for such models are intractable, and existing approximate solutions can be highly inaccurate. In this work we present a solution for linear switching state-space models that is able to recover the underlying hidden states and model parameters for time-varying dynamics in a way that is robust to model mis-specification and that outperforms previously proposed methods. We demonstrate the utility of our method by applying it to the problem of sleep spindle detection and show that switching state-space models can automatically detect transient spindle activity from human sleep electroencephalograms.

## Introduction

State-space modeling is a promising and versatile analytic approach for characterizing neural signals that has received substantial attention over many decades [1]. State-space models can be highly effective for analyzing neural time series data, particularly if the model formulation and parameterization are able to accurately represent the dynamics of the data generating process [2–4]. Powerful neural data analysis methods have been developed using stationary linear models [1, 5–9], for which inference and learning algorithms can be readily derived using well-established methods. However, real neural signal dynamics are more often time-varying, due to either intrinsic fluctuations of physiological states or extrinsic influences from cognitive and environmental processes [10–15]. Accordingly, mismatches between chosen state-space models and true underlying dynamical processes can be a major source of error in such algorithms [16].

In practice, researchers can manage variations in temporal dynamics by allowing the model parameters to vary across time or fitting different model parameters to consecutive windows of data. Thresholds can also be used to exclude outliers and limit the range of time series data within some neighborhood where a locally time-invariant approximation remains valid. These approaches can be effective when the time-varying activity can be tracked by changing parameter values within a given model class, or when the temporal dynamics evolve slowly compared to the window length. However, they fall short when the neural signals of interest are by definition transient. Examples of such transient activity are abundant in neuroscience, including epileptic bursts, hippocampal ripples, and event-related potentials [17–21]. In other scenarios, neural dynamics can change abruptly at unpredictable times, for instance, during anesthesia-induced burst-suppression, different stages of sleep, and modulations from active learning and behavioral responses [22–30]. Time-varying parameters or windowed approximations would perform sub-optimally at best in these cases, and could miss rapid transitions in the dynamics if the scale of temporal variation or window length is mis-specified.

State-space models with switching have been proposed to represent signals that are composed of multiple segments with different dynamics, under the assumption that each segment is approximately linear and stationary. Two distinct approaches have been taken to develop

corresponding algorithms for these switching state-space models. In one approach, approximate solutions were developed based on existing solutions [31] to stationary Gaussian state-space models, or Gaussian SSM for short. These early studies recognized the intractability of accounting for all possible switching transitions whose combinations grow exponentially with the length of the time series [32, 33]. Nevertheless, efficient inference algorithms have been developed by assuming that the transitions between multiple state-space dynamics follow a hidden Markov model (HMM) process and by approximating the conditional means and covariances of Gaussian hidden states [34–38]. These methods have focused on the filtered estimates of the switching model probabilities and therefore on the ability to forecast a transition in dynamics. While this inference is relevant in control systems and econometrics, data analysis questions in neuroscience are more often centered on segmenting a given time series into periods with distinct neural dynamics and subsequently characterizing the properties of those dynamics. Likely for this reason, these inference algorithms have not been extensively applied in neural signal processing.

In another more recent approach, advances in Bayesian methods have enabled the development of several new algorithms for switching state-space models [39–41]. These methods assume similar switching dynamics undergoing HMM transitions but leverage Markov chain Monte Carlo sampling techniques to perform state estimation and parameter learning. The adoption of blocked Gibbs sampling [41, 42] for switching state-space models is particularly powerful, as it allows extensions with non-parametric [39] and generalized linear models [42, 43] as well as with traditionally intractable structures such as recurrent dynamics [41]. These state-of-the-art methods have been successfully applied to the analysis of neural signals [43–47]. Empirical studies so far using these algorithms have employed less structured hidden state-spaces with dense transition matrices. A limitation of this general-purpose approach is that the candidate models may be less interpretable in relation to the neural mechanisms being studied compared to simpler and physiologically motivated models [48]. So far, the Bayesian sampling methods have been successfully applied to segmentation problems using high-dimensional embedding of neural signals [43–47]. Meanwhile, state-space models employing quasi-stationary sliding windows with closed-form solutions have been used to characterize time-varying signals [49–52]. The performance of such models in time-varying scenarios could benefit from extensions to incorporate switching dynamics.

In 2000, Ghahramani and Hinton proposed an insightful solution for switching state-space models inspired by variational approximations from graphical models [53]. Their generative models were similar to the hybrid models combining Gaussian SSMs and HMM, and they derived inference and learning computations in the recent Bayesian framework. But, instead of relying on Gibbs sampling, their approximate inference and learning solution utilized traditional exact inference algorithms as building blocks. The Ghahramani and Hinton algorithm is uniquely situated between the above two eras of research on switching state-space models, and it provides an opportunity to combine strengths from both approaches to study neural activity with transient dynamics using interpretable models and optimized approximations. This algorithm provides an accessible entry point to switching state-space models and facilitates construction of time-varying models of neural activity that could be further developed using more recent Bayesian methods. Despite the important and significant conceptual advance described by Ghahramani and Hinton, its applications in neuroscience have also been limited. Overall, the likelihood function for switching state-space models is non-convex and solutions are therefore sensitive to the initialization conditions. To address this issue, Ghahramani and Hinton used deterministic annealing, enabling the algorithm to perform comparably to past inference methods, but with little improvement. Moreover, the complexity of the algorithm and its computational requirements may have limited its adoption.

In this paper we introduce methods to significantly improve switching state-space model inference and learning under the framework initially proposed by Ghahramani and Hinton. First, we present a simple but rigorous derivation of the inference algorithm [53] under the variational Bayes approximation and expectation-maximization (EM) frameworks, complete with all closed-form equations involved along with practical considerations when applying this method to neuroscience problems. We then describe a novel initialization procedure that addresses the crucial challenge of navigating the non-concave log-likelihood function to find relevant optimal solutions. We show results on how this method significantly improves over deterministic annealing, enabling variational inference to outperform past inference algorithms. We also show that our approach improves parameter recovery, which makes it possible to more accurately characterize time-varying dynamics. We then extend the generative model structure to accommodate more complicated switching state-space models including nested models, which are frequently encountered in many applications including neuroscience. Finally, we apply the variational Bayesian learning algorithm to a long-standing problem of sleep spindle detection from electroencephalography (EEG) recordings during sleep in humans and show compelling results that switching state-space models can reliably identify transient neural activity.

## Results

Throughout this work, we use regular and boldface lowercase letters to denote scalars and vectors, respectively. Matrices are denoted by boldface uppercase letters or by regular uppercase letters when the matrices are one-dimensional. The transpose of a matrix $\mathbf{M}$ is denoted by $\mathbf{M}^\top$, and $\mathbf{M}_{ij}$ indicates the element of the matrix at the $i^{\text{th}}$ row and $j^{\text{th}}$ column position. A variable indexed with another discrete variable $s$ taking values in $\{1, \cdots, M\}$, e.g., $z^{(s)}$, refers to the following:

$$z^{(s)} = \begin{cases} z^{(1)} & s = 1 \\ z^{(2)} & s = 2 \\ \vdots \\ z^{(M)} & s = M. \end{cases}$$

We consider switching state-space models to consist of a set of $M$ uncorrelated linear Gaussian SSMs encoding arbitrarily distinct dynamics and an HMM of a discrete variable $s$ taking values in $\{1, \cdots, M\}$. The hidden states of Gaussian SSMs evolve in parallel and are allowed to be of different dimensions with appropriate mapping to observations. At every time point, the HMM selects one of the Gaussian SSMs to generate the observed data, giving rise to the switching behavior of this generative model. However, this flexible switching structure comes with high computational complexity: exact inference of the hidden states from observed data quickly becomes intractable, even for $M = 2$ with moderately long time series.

It has been noted that when the switching states are known, the hidden states of Gaussian SSMs can be efficiently estimated. Conversely, one can infer the hidden HMM states given the Gaussian SSM states [53]. Based on this insight, the intractability can be circumvented by using a surrogate distribution, $q$, which approximates the intractable posterior, $p$. Specifically, we introduce two auxiliary variables $g_t^{(m)}$ and $h_t^{(m)}$: $g_t^{(m)}$ acts as the model evidence for the $m^{\text{th}}$ Gaussian SSM to produce the observed data in the absence of known Gaussian SSM states, while $h_t^{(m)}$ represents the model responsibility for the $m^{\text{th}}$ Gaussian SSM to explain the observed data when the switching states are unknown. Therefore, alternately updating $g_t^{(m)}$ and

$h_t^{(m)}$ allows us to efficiently estimate posterior distributions of all hidden states in closed-form. The functional forms of these variables are obtained by maximizing a closeness metric between the two distributions $p$ and $q$. This procedure is known as variational approximation [54, 55]. We can also use this approximate variational inference within an instance of generalized EM algorithm to learn (fine-tune) the Gaussian SSM and HMM parameters when the model parameters are unknown (unsatisfactory). Fig 1 outlines this variational Bayesian learning algorithm for switching state-space models.

The variational inference procedure requires a *good* initialization of $g_t^{(m)}$ or $h_t^{(m)}$ so that they can be iteratively updated to drive the surrogate posterior $q$ closer to the intractable true posterior $p$ as described in Materials and methods. Given the non-convex nature of the problem, a *good* initialization should lead to a *good* local minimum. In practice, having an informed initialization is often difficult since no prior information on the discrete variable $s_t$ is available. One interesting quantity available from closed-form state inference is an *interpolated density*, defined as the conditional probability distribution of any particular observation, given all past and future observations. This interpolated density allows us to devise an informative initialization of the iterative variational inference procedure, instead of using deterministic annealing [53] (see details in section Initialization of fixed-point iterations of Materials and methods). Concretely, we use this density to compare between the Gaussian SSMs and establish the initial weights for the HMM, $g_t^{(m)}$, which enables us to achieve superior performance in both segmentation and parameter estimation.

In the following results sections, we first show simulation studies to assess the performance of such variational inference and learning. As performance metrics, we evaluate segmentation accuracy of a time series by learned switching states against the ground truth, and parameter estimation errors where applicable, to compare the proposed algorithm with a few existing switching inference algorithms. In addition, we investigate the effects of varying data length on segmentation and parameter learning metrics. Lastly, we model real-world human sleep EEG using switching state-space models and apply the proposed algorithm to detect the occurrence of sleep spindles in an unsupervised manner.

## Segmentation with posterior inference

We first focus on the variational inference part (E-step) of the algorithm that approximates the true posterior distribution $p$ with a structured approximate posterior distribution $q$ via fixed-point iterations (see the Variational approximation of hidden state posterior section of Materials and methods). To demonstrate improvements with our novel initialization procedure using the interpolated density, we repeated the simulations from [53] using the following switching autoregressive (AR) models of order 1:

$$
\begin{aligned}
x_t^{(1)} &= 0.99\, x_{t-1}^{(1)} + w_t^{(1)}, \quad w_t^{(1)} \sim \mathcal{N}(0, 1) \\
x_t^{(2)} &= 0.90\, x_{t-1}^{(2)} + w_t^{(2)}, \quad w_t^{(2)} \sim \mathcal{N}(0, 10) \\
y_t &= x_t^{(s_t)} + v_t, \qquad\quad\ v_t \sim \mathcal{N}(0, 0.1).
\end{aligned}
$$

The starting points for each AR model were drawn from the same distribution as their respective state noises. The switching state $s_t$ followed a binary HMM process with initial priors $\rho_1 = \rho_2 = 0.5$ and a symmetric state-transition probability matrix with $\phi_{11} = \phi_{22} = 0.95$, $\phi_{12} = \phi_{21} = 0.05$. We generated 200 sequences of 200 time points from the above generative model and analyzed the segmentation accuracy of inference algorithms given the true parameters.

Four inference algorithms were compared: static switching [33], interacting multiple models (IMM) [35], variational inference with deterministic annealing (VI-A) [53], and our

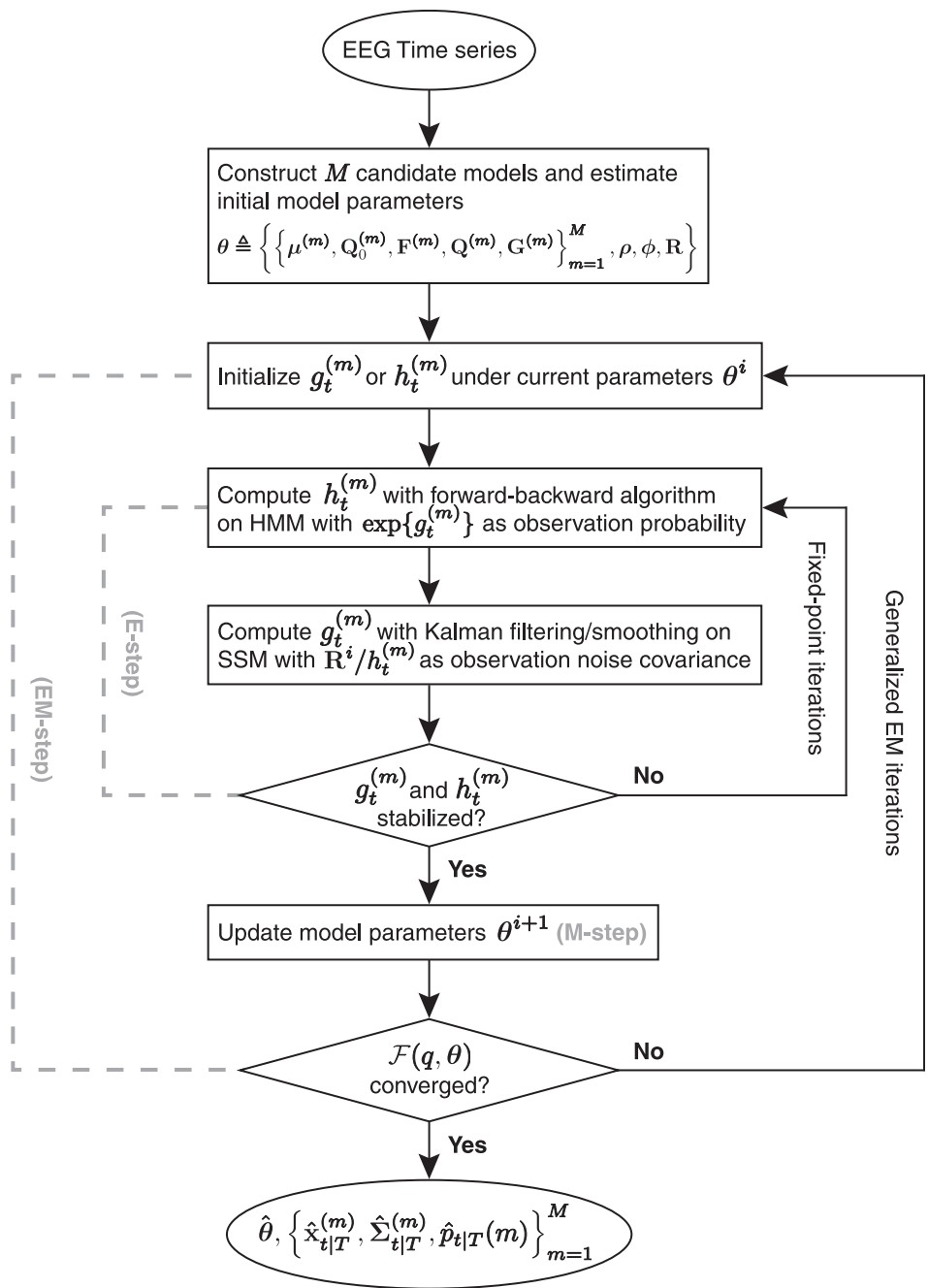

**Fig 1. Flowchart of the variational Bayesian learning as an instance of generalized EM algorithm.** $M$ Gaussian SSMs, indexed by $m \in \{1, \cdots, M\}$, is parameterized by $\left\{ \boldsymbol{\mu}^{(m)}, \mathbf{Q}_0^{(m)}, \mathbf{F}^{(m)}, \mathbf{Q}^{(m)}, \mathbf{G}^{(m)} \right\}$ and augmented with an HMM parameterized by $\boldsymbol{\rho}$ and $\boldsymbol{\phi}$. The HMM determines the switching among the Gaussian SSMs to produce observed data. The observations are corrupted by observation noise with covariance $\mathbf{R}$ and indexed by $t \in \{1, \cdots, T\}$. We introduce $g_t^{(m)}$ and $h_t^{(m)}$, two variational summary statistics, to approximate the true posterior distribution $p$ with a different distribution $q$. The E-step requires inference of the hidden states, achieved through fixed-point iterations that improve the variational approximation incrementally. Once the E-step has stabilized, model parameters are updated in the M-step. The convergence of the inner (E-step) iterations and outer (EM) iterations, index by $i$, is checked using the negative variational free energy $\mathcal{F}(q, \boldsymbol{\theta})$. This algorithm outputs 1) posterior estimates of model parameters, $\hat{\boldsymbol{\theta}}$, 2) state inferences, i.e., the means and covariances of Gaussian SSM hidden states, $\hat{\mathbf{x}}_{t|T}^{(m)}$ and $\hat{\boldsymbol{\Sigma}}_{t|T}^{(m)}$, and 3) estimates of $M$ model probabilities of generating the observation at each time point, $\hat{p}_{t|T}(m)$.

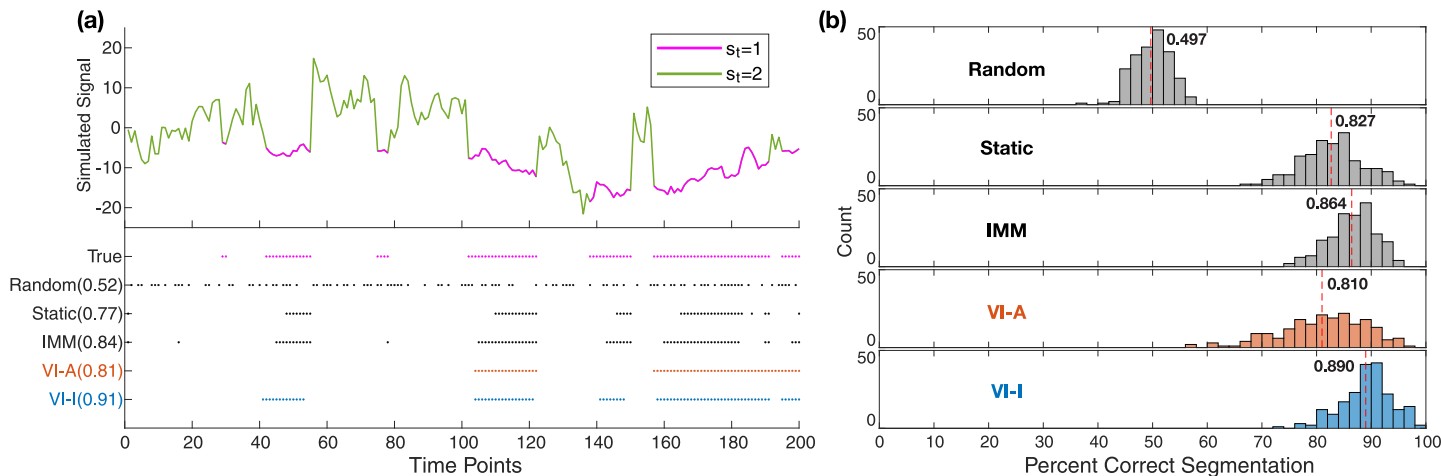

**Fig 2. Simulation results: Segmentation performance when true parameters are known.** (a) An example simulated sequence switching between two AR1 models with different dynamics. The top panel shows the time trace with the two states marked in different colors. The bottom panel shows switching inference results on this example given true parameters of the underlying generative model. Time points estimated to be in the first model ($s_t = 1$) are marked in colored dots for each inference method, with accuracy shown in parentheses. (b) Histograms of segmentation accuracy across 200 repetitions. The mean segmentation accuracy for each method is displayed and marked by the dashed red line. True = ground truth; Random = random segmentation with a Bernoulli process; Static = static switching method; IMM = interacting multiple models method; VI-A = variational inference with deterministic annealing (orange color); VI-I = variational inference with interpolated densities (blue color).

proposed variational inference with interpolated densities (VI-I). Static switching assumes the switching state $s_t$ to be independent across time points and applies the Bayes rule directly for switching inference [33]. IMM utilizes Gaussian merging to approximate the posterior distribution at each time point to estimate the switching state [35]. VI-A was initialized with a temperature parameter $\mathcal{T} = 100$ that decreased to $\mathcal{T} = 1$ over 12 fixed-point iterations with $\mathcal{T}_{i+1} = \frac{1}{2}\mathcal{T}_i + \frac{1}{2}$. For comparison, VI-I also ran for 12 fixed-point iterations. Switching states were labelled with a 0.5 threshold on the posterior model probabilities.

An example simulated $y$ is shown in Fig 2a along with the estimated switching state by random segmentation and the four inference algorithms. Histograms of percentage correct segmentation (Fig 2b) verify that the same results of VI-A are obtained as in [53] with mean accuracy 0.810. Surprisingly, both static switching and IMM were more accurate, with means at 0.827 and 0.864, respectively. VI-I achieved the best mean accuracy at 0.890, surpassing the other algorithms. Notably, in terms of the precision of correct segmentation, i.e., the width of the histograms in Fig 2b, all of static switching, IMM, and VI-I show superior performance over VI-A.

## Segmentation with parameter learning

An important advantage of the variational Bayesian framework over other approximate inference algorithms is the ability to learn model parameters using a generalized version of EM. When the underlying parameters are unknown, an accurate learning algorithm can improve segmentation performance over an arbitrary selection of parameters; conversely, if the learning algorithm is inaccurate, segmentation could become unreliable.

To investigate switching segmentation when the model parameters are unknown and must be learned from data, we conducted simulations similar to the study above using the following

generative model:

$$x_t^{(1)} = 0.90\, x_{t-1}^{(1)} + w_t^{(1)}, \quad w_t^{(1)} \sim \mathcal{N}(0, 2)$$
$$x_t^{(2)} = 0.70\, x_{t-1}^{(2)} + w_t^{(2)}, \quad w_t^{(2)} \sim \mathcal{N}(0, 10)$$
$$y_t = x_t^{(s_t)} + v_t, \qquad\quad v_t \sim \mathcal{N}(0, 0.1).$$

Initial state values followed their model-specific state noise distributions. An HMM process identical as before was used for the switching state $s_t$, and we again generated 200 sequences of 200 time points. We initialized all algorithms with random parameters drawn from uniform distributions centered around the true values for the $l^{\text{th}}$ sequence:

$$F^{(1)} = 0.90, \qquad F_l^{(1)} \sim \mathcal{U}_{[0.8, 1.0]}$$
$$F^{(2)} = 0.70, \qquad F_l^{(2)} \sim \mathcal{U}_{[0.6, 0.8]}$$
$$Q^{(1)} = 2, \qquad Q_l^{(1)} \sim \mathcal{U}_{[1,3]}$$
$$Q^{(2)} = 10, \qquad Q_l^{(2)} \sim \mathcal{U}_{[5,15]}$$
$$R = 0.1, \qquad R_l \sim \mathcal{U}_{[0.01, 0.2]}$$
$$\phi_{11} = \phi_{22} = 0.95, \quad \phi_{11,l} = \phi_{22,l} \sim \mathcal{U}_{[0.9, 0.99]}$$

where the notations for model parameters are described in the Real- and discrete-valued state-space models section of Materials and methods.

We compared the same four algorithms: for static switching and IMM inference algorithms, we estimated the segmentation of sequences using the randomly initialized parameters, since parameter learning is not part of these algorithms; for the VI-A EM and VI-I EM learning algorithms, we ran both the fixed-point iterations during the E-step and generalized EM iterations until convergence (see the Fixed-point iterations section of Materials and methods and S3 Appendix). The same temperature decaying in the variational inference analysis was used for the VI-A EM learning at every E-step. Switching states were labelled with a 0.5 threshold on the posterior model probabilities.

Percentage correct segmentation histograms show performance degradation as expected across all algorithms compared to inference with true parameters (Fig 3a). Despite the use of incorrect parameters, static switching and IMM achieved reasonable segmentation accuracy at 0.750 and 0.809, respectively, which decreased by 6–7% compared to the previous section where the true parameters were used. Notably, VI-A EM was much less accurate, with a mean accuracy of 0.705. In contrast, VI-I EM maintained excellent segmentation accuracy, outperforming the other algorithms, with a mean of 0.849 that only decreased 4.1% compared to the previous section. This pattern of better segmentation accuracy of VI-I EM over others was replicated when the same simulation study was repeated across a wide range of data length (see Fig 3c). Unlike static switching and IMM, both EM methods achieved greater segmentation accuracy as data length increased, followed by eventual plateauing. However, VI-I EM reached its plateau at a much higher accuracy than VI-A EM.

To characterize the robustness of variational learning and the ability to recover generative model parameters via EM, we analyzed the converged model parameters produced by the two variational learning algorithms relative to the true values (Fig 3b). Across model parameters, VI-A EM failed to identify the true values in most cases, suggesting the algorithm got trapped in local maxima of log-likelihood (see S4 Appendix for an analysis of negative free energy), which explains the poor segmentation accuracy in Fig 3a. VI-I EM estimated model

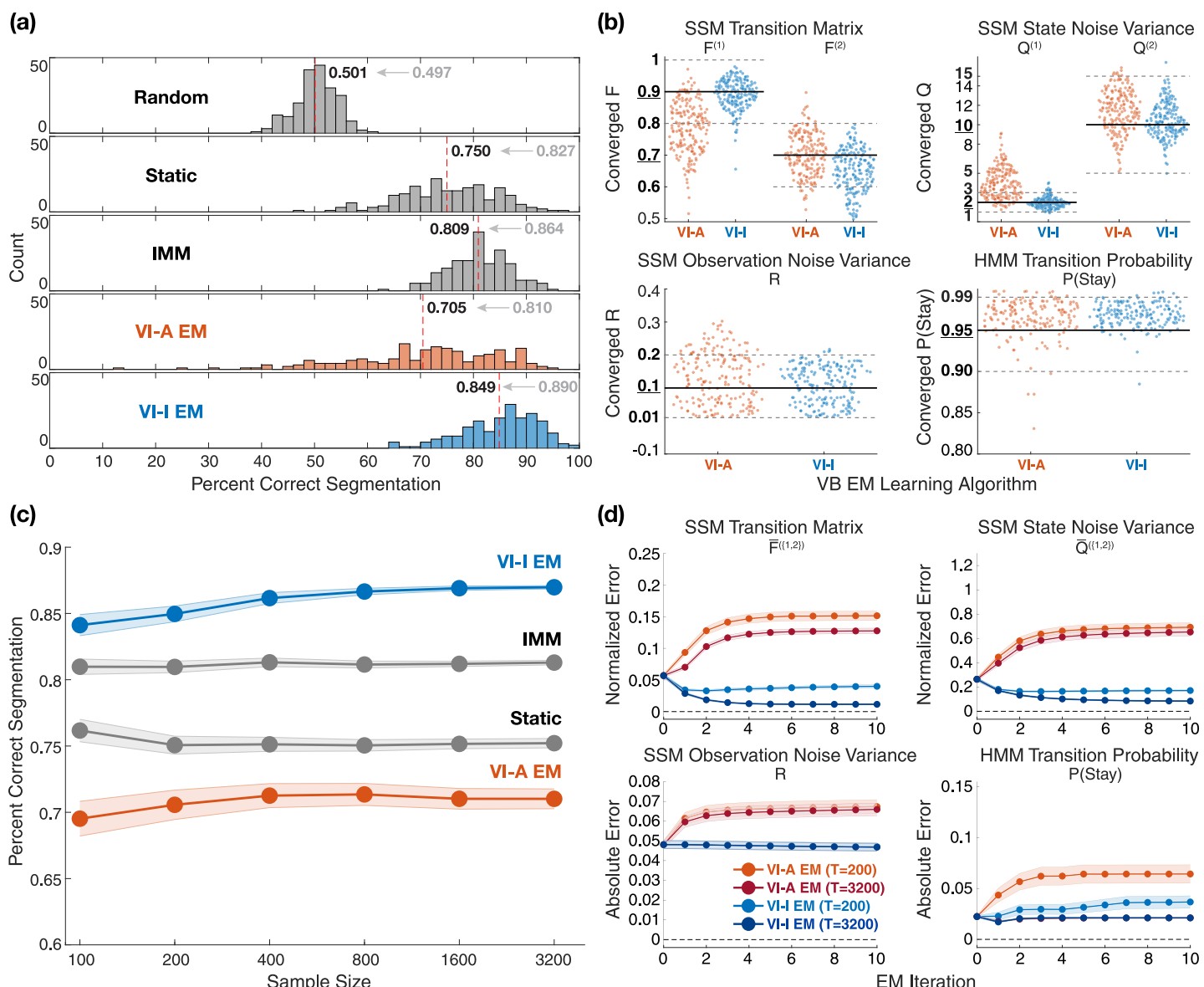

**Fig 3. Simulation results: Segmentation performance when true parameters are unknown.** (a) Histograms of segmentation accuracy across 200 repetitions of sequences of 200 time points. Since the true parameters were not known, they were estimated using an instance of EM algorithm for VI-A and VI-I inference starting from a random initialization. Static switching and IMM treated those random initialization as true model parameters. The mean segmentation accuracy for each method is displayed and marked by the dashed red line. The mean accuracy previously obtained using true parameters is shown in gray. Random = random segmentation with a Bernoulli process; Static = static switching method; IMM = interacting multiple models method; VI-A EM = variational EM learning with deterministic annealing (orange color); VI-I EM = variational EM learning with interpolated densities (blue color). (b) Swarm plots showing the distributions of model parameters learned by the variational learning algorithms for sequences of 200 time points. Uniform distributions used to sample initial parameters are marked in bold fonts on the y-axes, as well as using solid black lines (true values) and dotted gray lines (upper and lower bounds of the ranges). (c) Changes of mean segmentation accuracy over sequences of varying data lengths. Shaded bounds denote the standard error of the mean around the average accuracy values. (d) Mean parameter estimation errors from the true values across 10 EM iterations for two different data lengths. For transition matrix $F$ and state noise variance $Q$, normalized error is defined as abs(estimated—true)/true, and averaged across the two switching models. Absolute error is defined as abs(estimated—true). Shaded bounds denote the standard error of the mean around the average error values.

parameters that have distributional modes at the true values, indicating more successful parameter recovery than VI-A EM.

Tracking updated model parameters through 10 EM iterations reveals that estimated parameters converged quickly ($\sim$5 EM iterations) to their stationary values (Fig 3d). On average, the stationary values obtained by VI-I EM were closer to the true values than VI-A EM, which is consistent with Fig 3b. Additionally, some VI-A EM estimated parameters converged away from the true values. Increasing the data length reduced estimation errors in both methods while requiring slightly more EM iterations to reach desired convergence. The observation noise variance $R$ did not vary substantially during EM iterations, since its magnitude was much smaller compared to the state noise variances $Q^{\{1,2\}}$ (see S4 Appendix). Lastly, both methods tended to overestimate the probability of HMM to stay in the same model, a bias due to finite sampling that was attenuated for a longer data length (Fig 3d).

All of the simulations produced similar results when they were repeated with less informative distributions for model parameter initialization, albeit with slightly less accurate segmentation (see S4 Appendix). Overall, the variational learning method with interpolated densities showed robustness under uncertain model parameters.

### Extensions of model structure and parameter estimation

The simulations studied so far employ observations that switch between two AR1 models. While the findings are useful as a proof of principle for variational Bayesian learning, they are not representative of neural signals in real applications that require switching inference, as such signals usually have more complex dynamics. In addition, a binary HMM switching between two parallel processes introduces large change-point deviations as can be seen in Fig 2a, which might be trivially detected by looking at the first derivative of the time series. To address these gaps, we conducted simulation analyses using a single bivariate AR1 model that switches between two state-transition matrices. Specifically, the generative model is as follows:

$$\mathbf{x}_t = \begin{bmatrix} 0.5 & \mathbf{F}_{12}^{(s_t)} \\ 0 & 0.5 \end{bmatrix} \mathbf{x}_{t-1} + \mathbf{w}_t, \quad \mathbf{w}_t \sim \mathcal{N}\left(\mathbf{0}, \begin{bmatrix} 2 & 0 \\ 0 & 2 \end{bmatrix}\right)$$

$$\mathbf{y}_t = \begin{bmatrix} 1 & 0 \\ 0 & 1 \end{bmatrix} \mathbf{x}_t + \mathbf{v}_t, \quad \mathbf{v}_t \sim \mathcal{N}\left(\mathbf{0}, \begin{bmatrix} 0.1 & 0 \\ 0 & 0.1 \end{bmatrix}\right)$$

where $\mathbf{F}_{12}^{(1)} = 0.5$ and $\mathbf{F}_{12}^{(2)} = 0$. The switching state $s_t$ followed a binary HMM process as before with initial priors $\rho_1 = \rho_2 = 0.5$ and a state-transition probability matrix with $\phi_{11} = \phi_{22} = 0.95$, $\phi_{12} = \phi_{21} = 0.05$. Therefore, this generative model consists of two identical AR1 models where the second model has time-varying influence on the first. An example simulated time series $\mathbf{y}$ is shown in Fig 4 along with the resultant segmentation by the four inference algorithms using true parameters. We generated 200 sequences of 200 time points and analyzed the segmentation accuracy of inference algorithms given the true parameters. The same temperature parameters as before were used for variational inference with deterministic annealing. Both VI-A and VI-I ran until convergence of the fixed-point iterations. Switching states were labelled with a 0.5 threshold on the posterior model probabilities.

It is evident from both the generative model equations and the example time series that this is a more difficult inference problem than previous studies using two AR1 models with uncoupled dynamics. Nevertheless, all inference algorithms provided informative estimates compared to random segmentation, with the variational inference algorithms obtaining better accuracy results (Fig 5a). VI-I produced segmentation with the best accuracy mean at 0.741, followed by VI-A at 0.719. Static switching and IMM had worse performance at 0.634 and

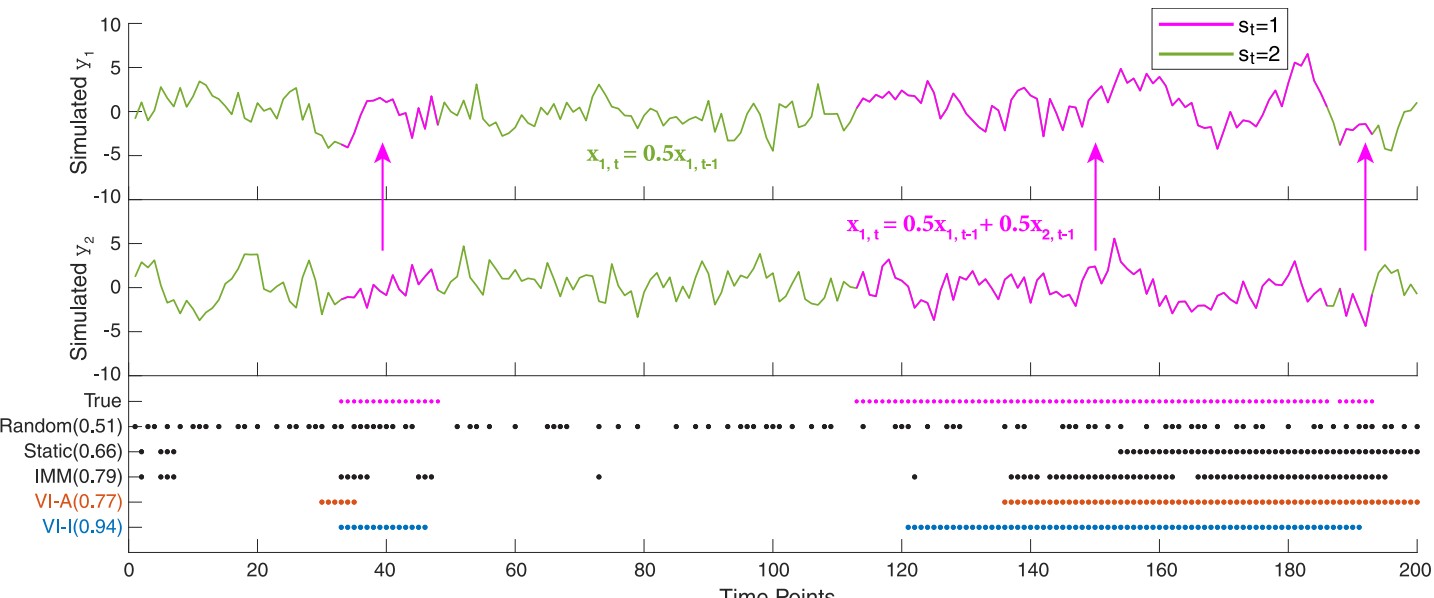

**Fig 4. Simulation results: Example segmentation on data generated from a different switching model class when true parameters are known.** The top two panels show the two sequences, $y_1$ and $y_2$, recorded as a bivariate observation **y**. Sequence $y_2$ has a non-zero influence on sequence $y_1$ as shown by the upward arrows, according to a switching state $s_t$. The time traces are also marked with different colors for the two switching states. The bottom panel shows the results of switching inference on the example data given true parameters. Time points estimated to be in the first model ($s_t = 1$) are marked in colored dots for each inference method, with accuracy shown in parentheses. True = ground truth; Random = random segmentation with a Bernoulli process; Static = static switching method; IMM = interacting multiple models method; VI-A = variational inference with deterministic annealing (orange color); VI-I = variational inference with interpolated densities (blue color).

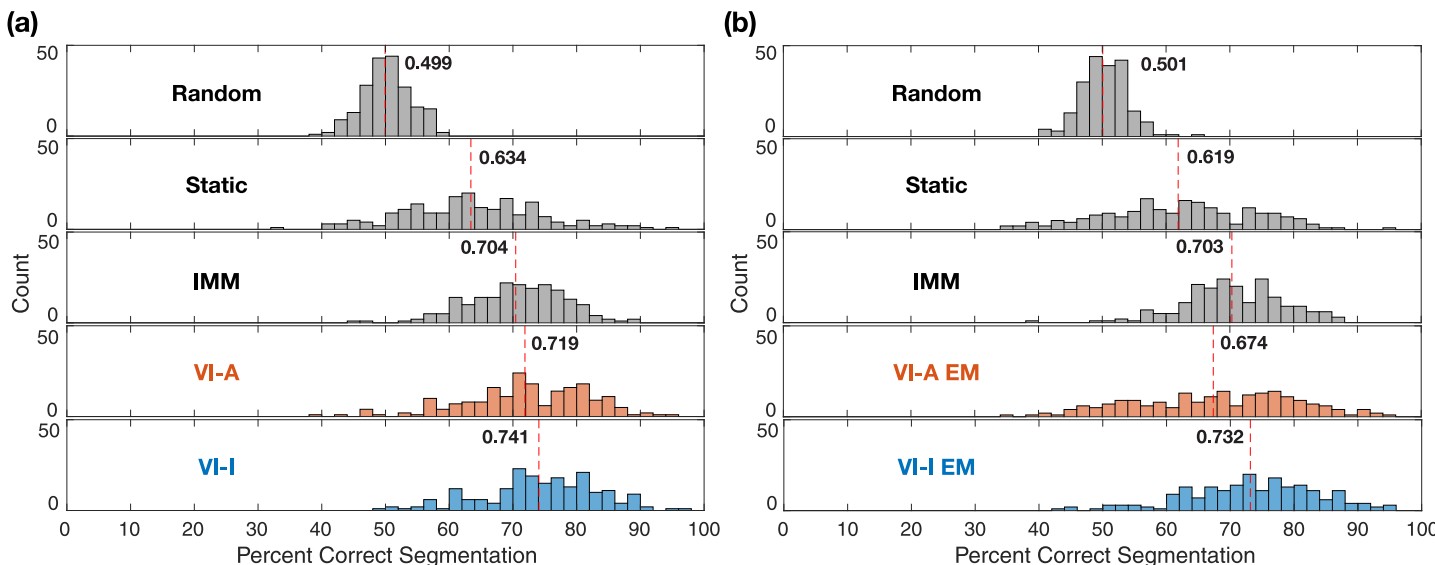

**Fig 5. Simulation results: Segmentation performance on data generated from a different switching model class.** (a) Histograms of segmentation accuracy given true parameters across 200 repetitions. (b) Histograms of segmentation accuracy when model parameter were unknown across 200 repetitions. We followed the same strategy stated in Fig 3 when the true model parameters were not available. In both (a) and (b), the mean segmentation accuracy for each method is displayed and marked by the dashed red line. Random = random segmentation with a Bernoulli process; Static = static switching method; IMM = interacting multiple models method; VI-A = variational inference with deterministic annealing (orange color); VI-I = variational inference with interpolated densities (blue color). VI-A/VI-I EM denote the EM learning algorithms with the corresponding initialization procedure during E-steps.

0.704 respectively. We note that the true generative model in this simulation is different from the assumed generative model employed during inference. Nonetheless, we see that VI-I still provides useful switching inference. This feature is discussed in more detail in the forthcoming section.

Next, we proceed to variational learning when model parameters are unknown. We initialized all algorithms for the $l^{\text{th}}$ sequence with parameters drawn from uniform distributions centered around the true values:

$$
\begin{aligned}
\mathbf{F}_{\{\backslash 21\}} &= 0.5, & \mathbf{F}_{11,l}, \mathbf{F}_{22,l}, \mathbf{F}_{12,l}^{(1)} &\sim \mathcal{U}_{[0.4,0.6]} \\
\mathbf{Q}_{11} = \mathbf{Q}_{22} &= 2, & \mathbf{Q}_{11,l}, \mathbf{Q}_{22,l} &\sim \mathcal{U}_{[1,3]} \\
\mathbf{R}_{11} = \mathbf{R}_{22} &= 0.1, & \mathbf{R}_{11,l}, \mathbf{R}_{22,l} &\sim \mathcal{U}_{[0.01,0.2]} \\
\boldsymbol{\phi}_{11} = \boldsymbol{\phi}_{22} &= 0.95, & \boldsymbol{\phi}_{11,l} = \boldsymbol{\phi}_{22,l} &\sim \mathcal{U}_{[0.9,0.99]}.
\end{aligned}
$$

Multivariate static switching and IMM were applied using the randomly initialized parameters. For VI-A EM and VI-I EM learning algorithms, we ran both the fixed-point iterations (E-step) and generalized EM iterations until convergence. Identical annealing and switching state decision thresholds were used as before.

With the M-step of variational learning in this analysis, we demonstrate an important extension of the structured variational approximation to support jointly estimated parameters across models. Specifically, the structured variational algorithms assume the following generative model structure in this bivariate AR1 example:

$$
\mathbf{x}_t^{(1)} = \mathbf{F}^{(1)} \mathbf{x}_{t-1}^{(1)} + \mathbf{w}_t^{(1)}, \quad \mathbf{w}_t^{(1)} \sim \mathcal{N}\left(\mathbf{0}, \mathbf{Q}^{(1)}\right)
$$

$$
\mathbf{x}_t^{(2)} = \mathbf{F}^{(2)} \mathbf{x}_{t-1}^{(2)} + \mathbf{w}_t^{(2)}, \quad \mathbf{w}_t^{(2)} \sim \mathcal{N}\left(\mathbf{0}, \mathbf{Q}^{(2)}\right)
$$

$$
\mathbf{y}_t = \begin{bmatrix} 1 & 0 \\ 0 & 1 \end{bmatrix} \mathbf{x}_t^{(s_t)} + \mathbf{v}_t, \quad \mathbf{v}_t \sim \mathcal{N}(\mathbf{0}, \mathbf{R})
$$

where the two candidate models have distinct transition matrices $\mathbf{F}^{(1)}$ and $\mathbf{F}^{(2)}$ that could be updated separately during the M-step, likewise for $\mathbf{Q}^{(1)}$ and $\mathbf{Q}^{(2)}$. However, the true generative model suggests that if two parallel models are used to approximate the switching dynamic, they should share all parameters except the $\mathbf{F}_{12}$ element of the state-transition matrix that gates the effect of the second sequence on the first sequence. Simple manipulations of the update equations can exploit this shared structure to reduce the set size of estimated parameters and pool information across candidate models. This joint parameter learning across models is described in detail in S2 Appendix.

As expected, the segmentation results in Fig 5b show slightly inferior performance compared to when using true parameters. The VI-I EM algorithm achieved the best mean accuracy at 0.732. Distributions of converged parameters were comparable between VI-I EM and VI-A EM (see S4 Appendix), suggesting better segmentation could still be obtained in the absence of obvious improvement in parameter recovery. In addition, when compared against a few different variational algorithms that are able to assume the accurate generative model structure using a single hidden state [41], VI-I EM also showed improved segmentation consistently (see S4 Appendix for more details).

This simulation study suggests that our method could be effective in analyzing neural signals that exhibit subtle switching dynamics, and in time-varying Granger causality problems particularly when the causal structure is changing rapidly [56–59]. In the current setting, the switching state can provide a statistical inference on the improvement of temporal forecasting

of the first sequence by the second sequence at every time point, without relying on log-likelihood ratio tests and windowing [60–63]. Our method also estimates parameters shared between the "full" and "reduced" models, while taking the observation noise into account [61, 64]. This application can be extended to higher-order models in a straightforward manner.

## Switching state-space oscillator models

The performance of methods based on parametric state-space models, such as the switching inference algorithms studied here, is ultimately dependent on how well the chosen model class represents the signal of interest. Oscillations are ubiquitous in neural signals, and therefore they need to be modeled with high sensitivity and specificity for the methods to capture any transient oscillations. Here, we employ a recently developed state-space model that is uniquely suited for modeling oscillations [65]:

$$\begin{bmatrix} x_{t,1} \\ x_{t,2} \end{bmatrix} = a \begin{bmatrix} \cos\omega & -\sin\omega \\ \sin\omega & \cos\omega \end{bmatrix} \begin{bmatrix} x_{t-1,1} \\ x_{t-1,2} \end{bmatrix} + \begin{bmatrix} w_{t,1} \\ w_{t,2} \end{bmatrix}, \quad \begin{bmatrix} w_{t,1} \\ w_{t,2} \end{bmatrix} \sim \mathcal{N}\left( \mathbf{0}, \begin{bmatrix} \sigma^2 & 0 \\ 0 & \sigma^2 \end{bmatrix} \right)$$

$$y_t = \begin{bmatrix} 1 & 0 \end{bmatrix} \begin{bmatrix} x_{t,1} \\ x_{t,2} \end{bmatrix} + v_t, \quad v_t \sim \mathcal{N}(0, R).$$

This model can be viewed as a phasor rotating around the origin with frequency $\omega$ (in radians) in the real and imaginary plane whose projection onto the real line produces the observed oscillation [66]. We refer to this model as an oscillator hereafter.

In many neuroscience applications, neural systems may exhibit different states defined by distinct combinations of oscillations, and the system may transition between these neural states over time. To test the applicability of our switching inference method for analyzing neural oscillations, we conducted a simulation study using hidden states composed of oscillators. We simulated up to 5 simultaneous oscillations at different frequencies with model parameters $a = 0.98$, $\sigma^2 = 3$ and $R = 1$ for 10 seconds under a sampling rate of 100 Hz. The frequencies were set at 1, 10, 20, 30, and 40 Hz respectively. With $n$ total underlying oscillations, one can generate $2^n - 1$ possible states, each with a different combination of the oscillations. We used a multinomial switching variable $s_t$ taking $2^n - 1$ values to select one of the states at each time point, to determine which oscillations were observed. The switching states followed an HMM process with uniform initial priors and a symmetric state-transition probability matrix with 0.98 on the diagonal. We repeated this data generation process 200 times for each of $n = 2, \cdots,$ 5 with the above oscillators. Fig 6a shows one such simulation instance where $n = 5$.

For example, if there are two oscillations with one at 1 Hz and the other at 10 Hz, there are three possible switching states, and the observation equation takes the form:

$$y_t = \mathbf{G}^{(s_t)} \mathbf{x}_t^{(s_t)} + v_t, \qquad v_t \sim \mathcal{N}(0, R)$$
$$\mathbf{G}^{(1)} = \begin{bmatrix} 1 & 0 \end{bmatrix}, \quad \mathbf{G}^{(2)} = \begin{bmatrix} 1 & 0 \end{bmatrix}, \quad \mathbf{G}^{(3)} = \begin{bmatrix} 1 & 0 & 1 & 0 \end{bmatrix}$$

where the hidden states $\mathbf{x}_t^{(1)}$ consist of only the 1 Hz oscillation, $\mathbf{x}_t^{(2)}$ of only the 10 Hz oscillation, and $\mathbf{x}_t^{(3)}$ of both oscillations. The switching state $s_t$ therefore takes values in {1, 2, 3}. We note that candidate models with more oscillators will inherently be favored by log-likelihood measures, owing to their higher numbers of degrees of freedom. To account for this, we mean-centered the model evidence of each candidate model when initializing the E-step, as described in detail in S3 Appendix.

We compared the segmentation accuracy of VI-A and VI-I given true parameters as we increased the number of underlying oscillations from 2 to 5. For simplicity, we considered the

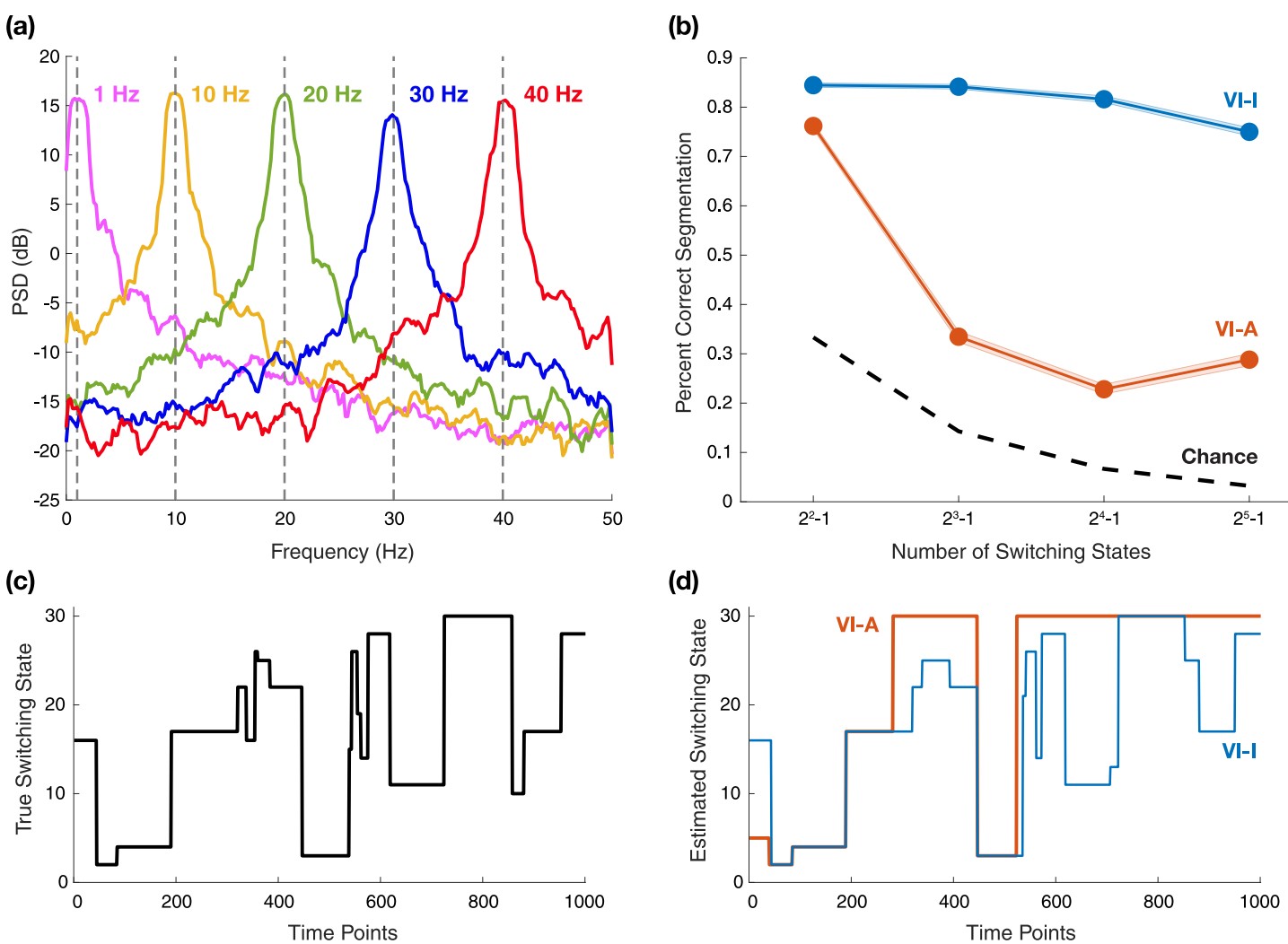

**Fig 6. Simulation results: Segmentation performance on switching state-space oscillator models when true model parameters are known.** (a) Power spectra of simulated oscillations at different frequencies; spectral estimation obtained via multitaper method that utilized 6 tapers corresponding to a time half-bandwidth product of 4. (b) Changes of mean segmentation accuracy over the number of switching states, as the number of underlying oscillations varies between 2 and 5. Shaded bounds denote the standard error of the mean around the average accuracy values across 200 repetitions. (c) An example switching state path with 5 underlying oscillations and 31 possible switching states. (d) Estimated switching states with variational inference. VI-A = variational inference with deterministic annealing (orange color); VI-I = variational inference with interpolated densities (blue color).

scenario where the model parameters are known, but as in previous examples, variational learning of the model parameters via EM is certainly possible. In practice, target neural oscillations are often characterized with stationary methods first; these models could be used as-is for inference, or to initialize model parameters for variational learning. Fig 6b shows that as the number of switching states increased, the accuracy of VI-A dropped precipitously while VI-I maintained good segmentation accuracy. VI-I was able to segment complicated switching trajectories close to the ground truth, such as in the example displayed in Fig 6c with 31 possible switching states from 5 oscillations, while VI-A failed to do so as shown in Fig 6d. These results demonstrate that our switching inference method is capable of modeling transient oscillations in a manner that scales well to higher-dimensional problems.

## Real-world application: Spindle detection

Sleep spindles are transient oscillatory bursts of neural activity observed in EEG recordings during non-rapid eye movement (NREM) sleep, with waxing and waning waveforms in the sigma frequency range (12 Hz–16 Hz) that last 0.5 s–3 s and that occur on a background of slow waves (0.5 Hz–2 Hz) [67]. In this section, we describe a novel spindle detection algorithm to illustrate how our developed switching method can be easily applied in the context of neural signal processing. As we will show, switching state-space models provide a rigorous and probabilistic framework for detection and extraction of spindles from sleep EEG recordings in an unsupervised way.

To model slow oscillations ($\delta$) and spindles ($\varsigma$) during sleep, we consider two independent state-space oscillators within a set of two candidate models:

$$\mathbf{x}_t^{(1)} = \begin{bmatrix} a^\delta \mathcal{R}(\delta) & \mathbf{0} \\ \mathbf{0} & a^\varsigma \mathcal{R}(\varsigma) \end{bmatrix} \mathbf{x}_{t-1}^{(1)} + \mathbf{w}_t^{(1)}, \quad \mathbf{w}_t^{(1)} \sim \mathcal{N}\left(\mathbf{0}, \begin{bmatrix} (\sigma^2)^\delta \mathbf{I}_2 & 0 \\ 0 & (\sigma^2)^\varsigma \mathbf{I}_2 \end{bmatrix}\right)$$

$$\mathbf{x}_t^{(2)} = a^\delta \mathcal{R}(\delta)\, \mathbf{x}_{t-1}^{(2)} + \mathbf{w}_t^{(2)}, \qquad\qquad \mathbf{w}_t^{(2)} \sim \mathcal{N}(\mathbf{0}, (\sigma^2)^\delta \mathbf{I}_2)$$

where $\mathbf{x}_t^{(1)} = \left[ x_{t,1}^{\delta^{(1)}}, x_{t,2}^{\delta^{(1)}}, x_{t,1}^{\varsigma^{(1)}}, x_{t,2}^{\varsigma^{(1)}} \right]^\top$ and $\mathbf{x}_t^{(2)} = \left[ x_{t,1}^{\delta^{(2)}}, x_{t,2}^{\delta^{(2)}} \right]^\top$; and

$$y_t = \mathbf{G}^{(s_t)} \mathbf{x}_t^{(s_t)} + \nu_t, \qquad \nu_t \sim \mathcal{N}(0, R)$$

$$\mathbf{G}^{(1)} = \begin{bmatrix} 1 & 0 & 1 & 0 \end{bmatrix}, \quad \mathbf{G}^{(2)} = \begin{bmatrix} 1 & 0 \end{bmatrix}$$

where $\mathbf{I}_2$ is an identity matrix, and for $f \in \{\delta, \varsigma\}$, $\mathcal{R}(f) = \begin{bmatrix} \cos \omega^f & -\sin \omega^f \\ \sin \omega^f & \cos \omega^f \end{bmatrix}$. An HMM process determines the switching state $s_t$ taking values in $\{1, 2\}$ across time. This model represents transient sleep spindles by switching between two "neural states": one with both slow oscillations and spindles, and the other with only slow oscillations. Furthermore, the slow ($\delta$) oscillator parameters are shared between the two candidate models to reflect the stationary dynamics of slow oscillations in the background.

To perform spindle segmentation, we initialized oscillator parameters as described in the Initialization of Gaussian SSM parameters section of Materials and methods. Briefly, the entire time series was modeled with a stationary state-space model including both slow oscillation and spindle oscillators, and parameters were learned through a standard EM algorithm. For variational learning, we derived M-step equations that considered the slow oscillators jointly across both candidate models, to update a single set of parameters for slow waves. In other words, the dynamics of the slow oscillations were modeled by pooling together segments regardless of the presence of sleep spindles. As we describe in S2 Appendix, this inference model structure can be viewed as close approximations to a range of other possible generative models for spindles. Lastly, the HMM switching process was initialized with initial state priors $\rho_1 = \rho_2 = 0.5$ and a state-transition probability matrix with $\phi_{11} = \phi_{22} = 0.99$, $\phi_{12} = \phi_{21} = 0.01$ that were updated during the M-steps. As mentioned in the earlier section, models with more hidden oscillation components will be favored by log-likelihood measures during learning. Thus, we again matched the ensemble averages of model evidence when initializing each E-step (see S3 Appendix). For VI-A EM, we used the same temperature decay as before.

We first compared variational learning methods with other inference algorithms on a short segment of EEG data recorded during sleep. In addition to static switching and IMM, we also analyzed a special Gaussian merging algorithm derived for Gaussian SSMs where only the observation matrix is allowed to switch [34]. Fig 7b shows that these inference-only algorithms performed poorly, even in this case with strong spindle activity: their posterior model

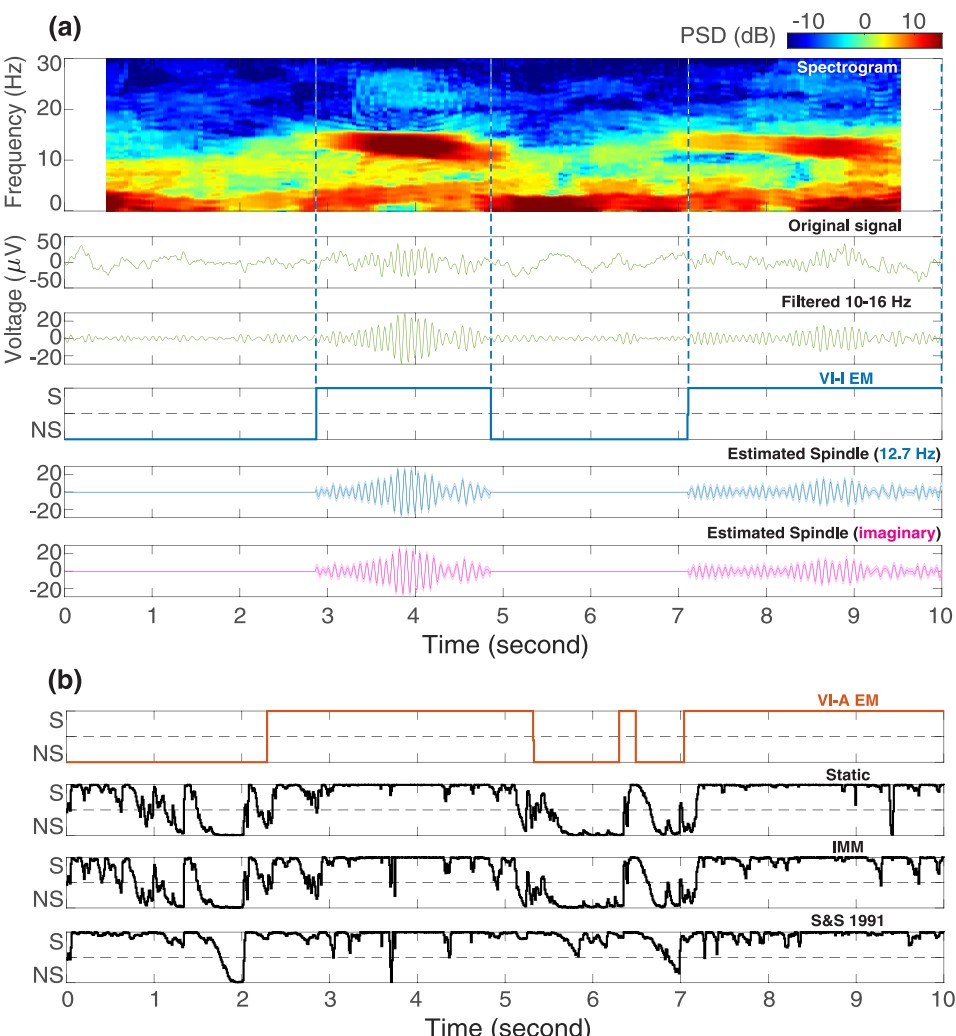

**Fig 7. Real-world example: Comparisons of switching inference algorithms for sleep spindle detection.** (a) In the top three panels, the spindle activity is visualized using a spectrogram, the original time trace, and after being bandpass filtered within 10 Hz–16 Hz. The fourth panel shows the posterior model probabilities of $s_t = 1$ estimated by variational EM learning with interpolated densities (VI-I EM, blue color). The margins of spindle events identified by VI-I EM are also marked with vertical dashed lines. The last two panels display the estimated real (blue) and imaginary (magenta) spindle waveforms with 95% confidence intervals from posterior covariances. The learned spindle center frequency is displayed in blue in parentheses. (b) Estimated posterior model probabilities of $s_t = 1$ by other algorithms in comparison. A model probability closer to 1 suggests the presence of spindles (S = Spindle), while being closer to 0 indicates no spindle (NS = No Spindle). The 0.5 model probability is marked with gray horizontal dashed lines. VI-A EM = variational EM learning with deterministic annealing (orange color); Static = static switching method; IMM = interacting multiple models method; S&S 1991 = the Gaussian merging method in [34].

probabilities were biased to select spindles due to the nested structure of the candidate models. On the other hand, VI-A EM labelled spindles that appeared reasonable given the EEG waveform and spectrogram. In comparison, VI-I EM produced segmentation results that were more accurate with tighter margins around spindles (Fig 7a). We note that the posterior model probabilities for VI-A EM and VI-I EM were more polarized (i.e., closer to either 0 or 1) compared to the other inference methods. This is a feature of the fixed-point iterations as described in Materials and methods, pushing the model responsibility $\mathbf{h}_t^{(m)}$ away from 0.5. We

show in S4 Appendix that a softer segmentation can be obtained using interpolated densities, while a harder segmentation is also possible via the Viterbi algorithm [68].

Besides estimating posterior model probabilities for spindle detection, this approach provides other parametric characterizations of spindle activity, such as the center frequency. For example, in Fig 7a, the algorithm learns that the spindles are centered around 12.7 Hz. In addition, the underlying spindle (and slow oscillation) waveform can be extracted via the inference procedure without any bandpass filtering (see Fig 7a). While the spindle waveform appears similar to the bandpass filtered signal, unlike bandpass filtering we can construct confidence intervals around the estimated spindle activity. We also emphasize that bandpass filtering requires a pre-defined frequency band, and this could introduce serious biases if the spindle frequency gets closer to the boundaries. Furthermore, the current method naturally limited spindle activity to the statistically relevant periods without arbitrary thresholding. Finally, these estimated hidden oscillator states (i.e., the real and imaginary parts) can be readily used to compute instantaneous amplitude and phase for downstream hypothesis testing [69, 70].

Next, we applied the best performing VI-I EM method to three randomly selected 30 s NREM stage 2 sleep recordings, robustly extracting sleep spindles across all time points (Fig 8). In addition, we compared the segmentation results to spindles identified by a wavelet-based automatic spindle detector [71], which has been shown to correspond well to human expert scoring [72]. Every spindle identified by the conventional method was also captured by VI-I EM. However, there were many other spindle events labelled by VI-I EM that were missed by the current "clinical standard". These events nevertheless showed obvious sigma-frequency power, suggesting a potentially inadequate detection by expert scoring. Furthermore, wavelet-based detection relied on arbitrary thresholds that only selected the strongest portion of a spindle activity. In contrast, actual spindles likely lasted for a longer duration, as evident both from the spindle waveforms estimated by VI-I EM and from the original signal time traces (Fig 8).

These results demonstrate that switching oscillator models with variational learning could detect spindles in an unsupervised manner that may be more reliable than expert scoring. In each recording, VI-I EM learned individualized parameters for the oscillators to best capture the time-varying amplitudes, frequencies, and waveform shapes of the underlying spindles. The posterior model probabilities of the HMM process provide a probabilistic criterion for spindle detection, which is not achievable with a simple fixed threshold on bandpass filtered data as in the de facto practice in sleep research.

## Discussion

In this paper we presented an algorithm for inference and learning with switching state-space models. This method holds promise in modeling time-varying dynamics of neural signals, in particular, neural time series data such as EEG. It takes a Variational Bayes approach to approximate the otherwise intractable posterior and enables state estimation and system identification via a generalized EM algorithm. We showed extensive simulation results on how the method can provide accurate segmentation of piece-wise linear dynamic regions, even when the true model parameters were unknown. We also extended the generative model structure to higher dimensions and to more subtle switching transitions instead of jumping between independent state-space models with distinct dynamics. Accordingly, we derived parameter learning rules that were modified to encode this structure and to provide more efficient parameter updates with joint estimation across switching models. Finally, we applied this learning algorithm to a real-world neural signal processing problem of sleep spindle detection, providing excellent detection and extraction of spindles with a rigorous statistical characterization.

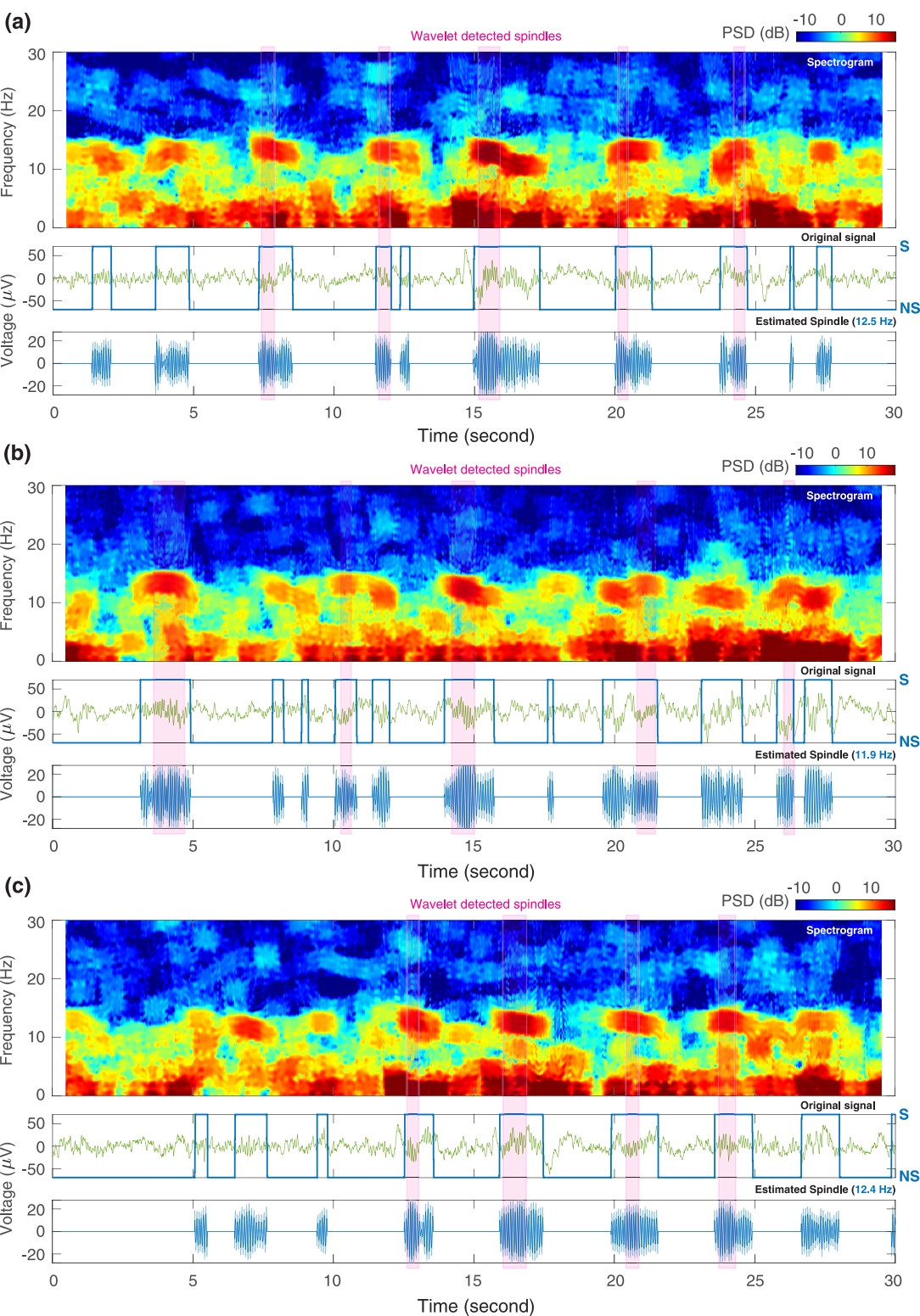

**Fig 8. Automatic segmentation of sleep spindles using the VI-I EM method.** Three 30 s EEG recordings of NREM-2 sleep were segmented with the variational EM learning method with interpolated densities (VI-I EM) to identify spindles in an unsupervised manner. In each of the (a)-(c) recordings, the three sub-panels visualize spindle activity using a spectrogram, the original time trace, and the estimated real part of spindle waveform with 95% confidence intervals from posterior covariances. The learned spindle center frequencies are displayed in blue in parentheses. The estimated posterior model probabilities for the

candidate model with both slow oscillations and spindles are overlaid on the time traces in blue lines. Shaded pink bars indicate spindles identified by a wavelet-based method for comparison. S = Spindle; NS = No Spindle.

Taken together, our results add to a growing literature on how switching state-space models can provide powerful statistical inferences to study neural signals.

We were inspired by an earlier, influential paper by Ghahramani and Hinton that applied the variational approximation to switching state-space models [53], but we introduced several important innovations that significantly improved the performance of the variational EM algorithm for these models in practice. In addition, we described in detail a simpler derivation of variational algorithms for such models. Ghahramani and Hinton utilized the same basic form of generative models presented here, but postulated the functional forms of the marginal distributions by introducing additional parameters $\mathbf{g}_t$ and $\mathbf{h}_t$ and solved for their optimal values by minimizing the KL divergence between the exact and approximate Hamiltonian functions. In contrast, we derived the optimal functional forms of hidden state marginals to maximize the negative variational free energy, which is mathematically equivalent to minimizing the KL divergence. In this way, we obtained expressions for $\mathbf{g}_t$ and $\mathbf{h}_t$ as summary statistics of the hidden state marginals directly from the variational posterior (see the Variational approximation of hidden state posterior section of Materials and methods).

Given the non-convexity of the problem, robust initialization of the generalized EM iterations, in particular of the fixed-point iterations during the E-step, is crucial for achieving reliable segmentation and avoiding trivial local maxima in the negative free energy. Ghahramani and Hinton proposed a deterministic annealing approach [53], which can be viewed as a form of variational tempering [73]. As discussed in [53] and analyzed in this paper, the annealed learning does not improve performance over traditional inference-only algorithms. We proposed a novel initialization strategy for $\mathbf{g}_t$, based on the insight that interpolated densities provide a local statistical comparison between the candidate models, agnostic of switching dynamics. The interpolated densities therefore provide natural starting values for $\mathbf{g}_t$, which serves as the surrogate observation probabilities in the HMM switching process. Our results showed that this technique significantly improved the segmentation accuracy and system identification of our algorithm compared to the deterministic annealing approach. We found that this more informed initialization was essential in more complicated scenarios, such as time-varying bivariate AR processes and switching oscillator models, or in practical neural signal processing applications, such as sleep spindle segmentation.

This algorithm provides a unique take on inference and learning for switching state-space models. Compared to traditional inference-only algorithms, also known as assumed density filtering (ADF) [74], the variational learning method allows iterative parameter tuning that improves segmentation. These learned parameters also allow us to characterize the properties of the underlying physical system, which may be important in scientific applications such as neural signal processing. Recent algorithms have explored smoothed inference by extending ADF algorithms [75–77]. In contrast, our approach provides an elegant solution for smoothed inference using familiar message-passing algorithms such as the Kalman filter, RTS smoother, and forward-backward algorithm, which are applicable after the conditional dependencies between the real- and discrete-valued hidden states are ignored under the parallel-model variational approximation [53]. This approach is an instance of structured variational inference for time series data [55].

Other variational learning methods have been developed for switching linear dynamical systems [78–80] based on the same principle of approximating intractable posterior with

factorized distributions. A notable difference is that these methods assume a single multi-modal hidden state that is modulated by the switching state instead of the multiple parallel state-space models in our generative process. As we explored in the bivariate AR1 simulation and spindle analyses, multiple parallel models can be an effective way to approximate multi-modality: the hidden states in parallel models are suppressed when they are not contributing to the observed data, since the Kalman filtering and smoothing within each model are weighed by the model responsibility $\mathbf{h}_t$. In addition, when multiple models contain hidden states reflected in the observed data, the smoothed estimates across parallel models are closely linked since they are all conditioned on the same observed data. A future study could compare these two approaches and characterize their computational costs and learning performance. Nevertheless, our initialization strategy using interpolated densities could still be helpful in these multi-modal variational learning algorithms. Given that the observed data have a sufficiently high sampling rate relative to the underlying state-transition dynamics, interpolated densities can provide local comparisons among switching states to warm start the E-steps of variational learning algorithms close to the desired optimal solution.

More recent algorithms for switching state-space models embrace a Bayesian approach by using sampling techniques to obtain posterior distributions. Some of these methods can simultaneously learn the number of switching states [39] and more complicated recurrent structures [41]. Recurrent neural networks can also be adapted to support Kalman updates in a deep state-space model and to characterize switching dynamics after training [81]. These methods have varying computational cost and complexity, which require careful adaptation for neural signal processing or biomedical applications. A few studies have applied variational inference on more elaborate model structures and showed meaningful segmentation with neural signals [43–47].

In comparison to the existing applications of switching state-space models, we focus on interpretable linear Gaussian SSMs that can faithfully capture the distinct dynamics observed in neural signals. By applying an intuitive variational approximation on several candidate models constructed from clinical and neuroscience domain knowledge, the variational learning method we described here provides a general solution for segmenting a given time series into neural dynamic states of interest. This method, however, still has limitations encountered in other switching state-space modeling algorithms. First, many of the methodological considerations we described here, including the bias of log-likelihoods in nested structures, apply to other methods. In addition, a single phenomenon or physical process may have multiple switching state-space representations. For example, the process described in the univariate AR1 simulations can have an alternative generative process: two univariate hidden states may be concatenated into a single bivariate hidden state with fixed dynamics, while an observation matrix gets modulated by the switching state to choose particular indices of the bivariate hidden state. Parameter learning and interpretation can be difficult in this scenario if the parameters for the SSM dynamics and the corresponding observation matrices are both learned from the observed data. Thus, we opted to use fixed observation matrices as a constraint to facilitate recovering unique solutions. We nonetheless provide expressions to update the observation matrices in Eq 28, as in some neuroscience applications, switching observation matrices may best encode neural dynamics of interest. In this scenario, they can be learned in a data-driven way with our approach after sharing all SSM parameters across candidate models, analogous to constraining to a stationary SSM dynamic in a single multi-modal hidden state [75].

Our assumed generative structure, consisting of multiple independent dynamical models, permits efficient closed-form computations on the approximate posterior using existing state-space inference algorithms. While the fixed-point iterations and parameter updates can be rigorously derived and familiar methods can be easily plugged in, the non-concave negative free

energy poses challenges for inference, including frequent local maxima with polarized model probability estimates that we report here. An alternative to the generalized EM algorithm is blocked Gibbs sampling, which has been successfully applied to switching state-space models [39–41]. A future direction for further development could be to apply blocked Gibbs sampling [82] to the class of neural oscillator models, in a manner that uses the interpolated density to improve computational efficiency.

In the same vein of simplifying the state estimation using using sampling techniques, black-box variational inference is also applicable to switching state-space models [83–85]. Black-box variational approximations are attractive because they provide a general-purpose tool requiring almost no distributional assumptions on the dynamical systems. However, they might not achieve as good performance in cases where we can identify candidate models and their switching structure from prior domain knowledge (see S4 Appendix for an example with the bivariate AR1 simulations). In addition, closed-form inferences tend to have greater computational efficiency than sampling techniques despite the requirement for fixed-point iterations, which converge quickly when the parallel models are reasonably initialized. One clear advantage of sampling techniques is that they are readily applicable in the case of non-linear observation models, such as observations generated by counting processes (e.g., point processes to model neural spiking activity with history dependence [86]). In this context, we note that our approach can also be extended to such non-linear observation models with relatively small effort. For example, point process observations can be handled by replacing Kalman filtering with extended Kalman filtering [87] or unscented Kalman filtering [88], with the rest of the switching inference procedure unaltered.

In general, the variational inference algorithm described here is best suited for scenarios where an informed guess of model parameters can be used for initialization. From a neural signal processing perspective, we are equally concerned about characterizing interpretable hidden dynamics of different neural states and achieving accurate segmentation of time-varying activity due to transitions among them. This intention is reflected in our use of multiple models in parallel, which compete with each other to explain observed data, rather than using a larger array of models with arbitrary structures and parameterizations that are intended to approximate neural dynamics of interest. In contrast, the parallel models we use favor a more parsimonious and informed set of candidate models to obtain interpretable characterization of dynamics and accurate segmentation. The use of the interpolated density is consistent with this premise, since it better exploits prior domain knowledge and the carefully constructed candidate models. In cases where there is an unknown number of switching states, the dynamics are less well-characterized, or the goal is to fit to data as well as possible, a number of sampling-based algorithms such as in [39, 41, 42, 44] could be highly effective.

The sleep spindles analyzed here serve as a good example of scenarios where switching state-space models can be directly applied to achieve much more powerful analyses than standard methods employed in the field. Spindles have been studied with visual scoring by trained human experts for decades [89], until recent efforts to develop reliable detection algorithms [90] in light of their integral roles in normal sleep physiology, memory consolidation, and psychiatric and neurodegenerative diseases [91]. While different in the exact implementation, almost all spindle detection algorithms are fundamentally based on setting an arbitrary threshold and picking strong transient signals in predefined frequency ranges [72], which is severely biased as shown in a recent study [92]. With informed prior knowledge of sleep oscillations, state-space oscillator models can be constructed. The exact parameters of these models should be learned from data since they vary across individuals and different stages of sleep. But one can decide on the number of switching models and their compositions given domain knowledge of sleep and the target spindle detection problem. In this study, we show that the

variational learning method can robustly extract sleep spindles from NREM2 sleep in an unsupervised manner, which obviates the need for thresholding. At the same time, variational learning can better characterize spindle properties such as frequency and amplitude by focusing on periods when they are statistically detectable. As mentioned earlier, the use of state-space oscillator models allows one to directly estimate instantaneous amplitude, phase, and frequency without relying on traditional bandpass filtering and Hilbert transform [69], as well as to easily obtain derived measures such as phase–amplitude coupling [70]. In addition, the state-space formulation also allows for filtered estimates of posterior switching state probabilities for online detection of sleep spindles, making it potentially suitable for real-time applications (see S4 Appendix for an example). Thus, there are numerous analytical advantages to a switching state-space modeling approach to detect and extract spindles.

Similar applications can be widely found across neuroscience: burst-suppression induced by anesthetic drugs, epileptic bursts, event-related potentials during cognitive tasks, spike train recordings, or simply behavioral responses over a block of trials [1]. Switching state-space models provide a statistical framework to investigate temporal variations and extract distinct states that are ubiquitous in different datasets. Thus, we advocate for a broad adoption of switching methods in place of previous windowing techniques that limit the analysis of neural signals.

## Conclusion

In this paper, we explored the use of switching state-space models for neural signal processing. We demonstrated the promising performance of a variational learning algorithm with interpolated densities, both in simulations and when applied to spindle detection in real sleep EEG data. Our method can be widely applied to neuroscience problems when the observed dynamics show transitions between periods that can be modeled by linear Gaussian state-space models.

## Materials and methods

### Definitions

We denote a sequence of values or vectors defined for $t = (0), 1, \cdots, T$ as $\{\bullet_t\}$, with $\{\bullet_t\}_a^b$ indicating $t = a, \cdots, b$. We denote the matrix trace operator by $\mathrm{Tr}\{\bullet\}$. Finally, by $\mathbf{z}_t \sim \mathcal{N}(\boldsymbol{\mu}, \boldsymbol{\Sigma})$, we mean the Gaussian distribution:

$$p(z_t) = |2\pi\boldsymbol{\Sigma}|^{-1/2}\exp\left\{-\frac{1}{2}(\mathbf{z}_t - \boldsymbol{\mu})^\top \boldsymbol{\Sigma}^{-1}(\mathbf{z}_t - \boldsymbol{\mu})\right\} \tag{1}$$

**Real- and discrete-valued state-space models.** A state-space model (SSM) defines a probability density of real-valued time series data, $\{\mathbf{y}_t\}$, by relating them to a sequence of hidden state vectors, $\{\mathbf{x}_t\}$, which follow the Markov independence property so that the joint probability for the state sequence and observations can be factored as:

$$p(\{\mathbf{y}_t, \mathbf{x}_t\}) = p(\mathbf{x}_0)\prod_{t=1}^{T} p(\mathbf{x}_t|\mathbf{x}_{t-1})p(\mathbf{y}_t|\mathbf{x}_t) \tag{2}$$

In its simplest form, the transition and output processes are linear and time invariant, incorporating multivariate Gaussian uncertainties [93]:

$$\mathbf{x}_t = \mathbf{F}\mathbf{x}_{t-1} + \mathbf{w}_t, \quad \mathbf{w}_t \sim \mathcal{N}(\mathbf{0}, \mathbf{Q}) \tag{3}$$

$$\mathbf{y}_t = \mathbf{G}\mathbf{x}_t + \mathbf{v}_t, \quad \mathbf{v}_t \sim \mathcal{N}(\mathbf{0}, \mathbf{R}) \tag{4}$$

where **F**, **G** are called state-transition and observation matrices. Provided $p(\mathbf{x}_0)$ is Gaussian, this set of equations defines a joint Gaussian density on $\{\mathbf{y}_t, \mathbf{x}_t\}$—we use the term Gaussian SSM to denote a model of this form.

A closely related type of state-space models has hidden states, $\{s_t\}$, which are discrete-valued, i.e., a multinomial variable that can take one of $K$ values. Using the Markov independence property, the joint probability for the discrete-valued state sequence and observations can be again factored as in Eq 2, with $s_t$ taking place of $\mathbf{x}_t$:

$$p(\{\mathbf{y}_t, s_t\}) = p(s_0) \prod_{t=1}^{T} p(s_t|s_{t-1}) p(\mathbf{y}_t|s_t) \tag{5}$$

The state-transition probabilities, $p(s_t|s_{t-1})$, are specified by a $K \times K$ state-transition matrix, $\boldsymbol{\phi}$. If the observed data are also discrete symbols taking one of L values, the observation probabilities can be fully specified by a $K \times L$ observation matrix, $\boldsymbol{\psi}$. This type of models is known as hidden Markov models (HMM) [94–96].

Next, we outline the various existing exact inference and learning algorithms associated with Gaussian SSM and HMM, in order to define the terminologies used throughout this paper. These algorithms will serve as the building blocks for inference and learning computations of switching state-space models described later.

**State estimation.** Given the observations and model parameters, inferring the hidden states of SSM has well-established solutions in the literature for linear Gaussian SSM with parameters $\{\mathbf{F}, \mathbf{Q}, \mathbf{G}, \mathbf{R}\}$ and for HMM with parameters $\{\boldsymbol{\phi}, \boldsymbol{\psi}\}$.

For Gaussian SSM, the joint distribution of the observations and hidden states given in Eq 2 is Gaussian, so inference on the hidden states also induces Gaussian posteriors. Various different application scenarios call for three kinds of inference problems: filtering, smoothing, and prediction [97]. *Filtering* deals with computing the posterior distribution $p\left(\mathbf{x}_t|\{\mathbf{y}_t\}_1^t\right)$ of the current hidden state, $\mathbf{x}_t$, conditioned on the observations up to time $t$. The recursive algorithm that carries out the computation is known as Kalman-Bucy filter [31]. The *smoothing* problem addresses finding the posterior probabilities $p\left(\mathbf{x}_t|\{\mathbf{y}_t\}_1^T\right)$ of the hidden states $\mathbf{x}_t$ given also future observations, i.e., up to time $T > t$. A similar recursive algorithm running backward from $T$ to $t$ implements the computation [98]. This backward recursion, combined with the Kalman filter running forward recursions from time 1 to $t$, is called Rauch-Tung-Striebel (RTS) smoother [99]. Finally, *prediction* computes the posterior predictive distribution $p\left(\mathbf{x}_{t+\tau}|\{\mathbf{y}_t\}_1^t\right)$ of the future hidden states, $\mathbf{x}_{t+\tau}$, conditioned on observations up to time $t$, as well as $p\left(\mathbf{y}_{t+\tau}|\{\mathbf{y}_t\}_1^t\right)$ given the observation matrix that relates future hidden states to observations.

For HMM, there are two similar inference problems of interest [94]. First, the recursive forward-backward algorithm computes the posterior probabilities of the discrete hidden states given observations from time 1 to $T$. As the name suggests, the forward pass computation steps are analogous to the Kalman filter, while the backward pass steps are analogous to the RTS smoother. The second form of inference deals with decoding the most likely sequence of hidden states that could generate the observations. A well-known solution is given by the Viterbi algorithm that relies on similar forward and backward passes through all time points [68].

**System identification.** The problem of finding the model parameters $\{\mathbf{F}, \mathbf{Q}, \mathbf{G}, \mathbf{R}\}$ and $\{\boldsymbol{\phi}, \boldsymbol{\psi}\}$ is known as system identification in the engineering literature. In the most general form, these problems assume that only the observed data sequence is accessible. Given the probabilistic nature of the models, one can choose to either seek a single locally optimal point estimate of the parameters (Maximum likelihood (ML) or Maximum a posteriori (MAP) learning) or follow

a Bayesian approach to compute or approximate posterior distributions of the model parameters given the data. In the system identification context, estimates from the former category (i.e., ML or MAP learning) are more popular and make use of the EM algorithm [100]. The EM algorithm alternates between optimizing the posterior distribution of the hidden states given current parameter estimates (the E-step) and updating the parameter estimates from the optimized posterior distribution of the hidden states (the M-step). This general procedure is guaranteed to increase the log-likelihood of the observed data sequence w.r.t. the model parameters [100].

For Gaussian SSM, the E-step is realized by the RTS smoother, and the M-step takes the form of a linear regression problem [93]. For M-steps with ML estimates, the linear regression problem remains unconstrained. On the other hand, for M-steps with MAP estimates, priors on the model parameters put constraints on the log-likelihood. Update equations can be derived in closed-form after taking derivatives with respect to each of the model parameters [93].

In the case of HMM parameters, the E-step utilizes the forward-backward algorithm to infer the posterior probabilities of the hidden states [94]. The M-step uses the expected counts of the discrete-valued state transitions and observations to update the state-transition and observation matrices with ML or MAP estimates. This procedure, also known as the Baum-Welch algorithm [101], predates the EM algorithm above.

## Switching state-space models

We employ one particular form of switching state-space models that allows time-varying dynamics among arbitrary real-valued state-space models with stationary parameters [53]. In addition to its more general formulation compared to modeling time-varying parameters within a single model, this construction offers an elegant solution under the variational Bayesian framework [54] as detailed later. The generative model consists of $M$ linear Gaussian SSMs indexed by numbers from 1 to $M$ which each contain continuous real-valued states, and one HMM whose states take on discrete integer values from 1 to $M$. Furthermore, the states within each of the Gaussian SSMs are assumed to evolve independently from other models. The discrete HMM too is independent of the Gaussian SSM and decides which one of the $M$ state-space models is generating the current observation data point. The directed acyclic graph corresponding to this conditional independence relation is shown in Fig 9.

The real-valued hidden states in the $M$ Gaussian SSMs are labelled as $\left\{ \mathbf{x}_t^{(m)} \right\}$ with $m \in \{1, \cdots, M\}$, the discrete-valued hidden states of HMM as $\{s_t\}$ where $s_t \in \{1, \cdots, M\}$, and the real-valued observations as $\{\mathbf{y}_t\}$. The joint probability for the observations and hidden states therefore factors nicely as:

$$
\begin{aligned}
p\left( \left\{ \mathbf{y}_t, \mathbf{x}_t^{(1)}, \cdots, \mathbf{x}_t^{(M)}, s_t \right\} \right) = \quad & \prod_{t=1}^{T} p\left( \mathbf{y}_t | \mathbf{x}_t^{(1)}, \cdots, \mathbf{x}_t^{(M)}, s_t \right) \\
& \times \prod_{m=1}^{M} \left( p\left( \mathbf{x}_0^{(m)} \right) \prod_{t=1}^{T} p\left( \mathbf{x}_t^{(m)} | \mathbf{x}_{t-1}^{(m)} \right) \right) \\
& \times p\left( s_0 \right) \prod_{t=1}^{T} p(s_t | s_{t-1})
\end{aligned}
\tag{6}
$$

where all three parts have the familiar forms from classical Gaussian SSM and HMM literature. We provide each of these expressions below.

First of all, conditioned on the discrete hidden state, $s_t = m$, the observation is simply a multivariate Gaussian variable from the observation equation of the $m^{\text{th}}$ Gaussian SSM. The

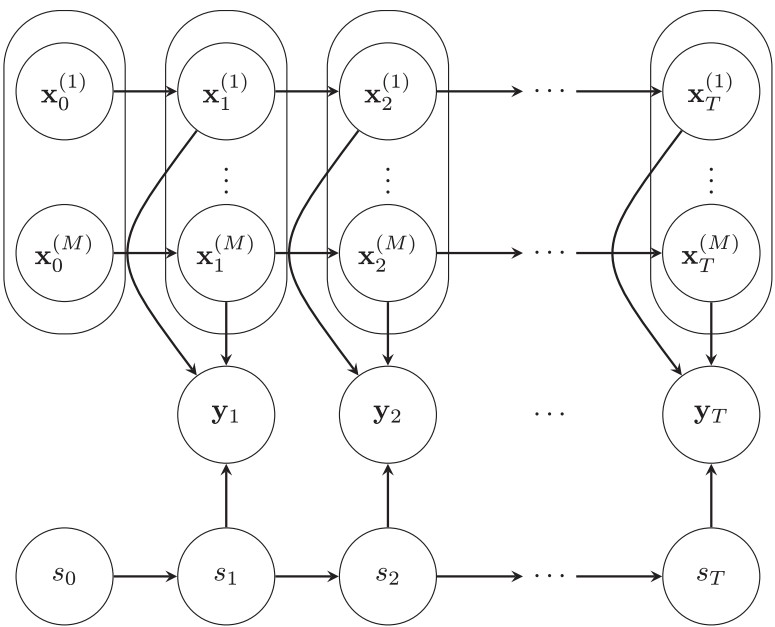

**Fig 9. Generative structure with parallel switching state-space models.** A directed acyclic graph is shown to represent the conditional independence structure between the $M$ real-valued Gaussian hidden state sequences $\{\mathbf{x}_t^{(m)}\}$ for $m \in \{1, \cdots M\}$, a discrete-valued hidden Markov chain $\{s_t\}$, and the observed data $\{\mathbf{y}_t\}$ up to time $T$. In this generative structure, the observation at a given time point depends only on the hidden states of the $M$ Gaussian models at that point, with the discrete-valued state selecting one of the models to produce the observation, hence the name *switching*.

probability distribution of the observation vector is given by:

$$p\left(\mathbf{y}_t | \mathbf{x}_t^{(1)}, \cdots, \mathbf{x}_t^{(M)}, s_t = m\right) =$$

$$|2\pi\mathbf{R}|^{-1/2}\exp\left\{ -\frac{1}{2}\left(\mathbf{y}_t - \mathbf{G}^{(m)}\mathbf{x}_t^{(m)}\right)^{\top}\mathbf{R}^{-1}\left(\mathbf{y}_t - \mathbf{G}^{(m)}\mathbf{x}_t^{(m)}\right)\right\} \tag{7}$$

where $\mathbf{R}$ is the observation noise covariance matrix, and $\mathbf{G}^{(m)}$ is the observation matrix of the linear Gaussian SSMs indexed by $m$.

Secondly, the real-valued states of the $M$ Gaussian SSMs evolve independently and in parallel, with dynamics specified by the model-specific state-transition matrix, $\mathbf{F}^{(m)}$, and state noise covariance, $\mathbf{Q}^{(m)}$, starting from different initial states, $\mathcal{N}\left(\boldsymbol{\mu}^{(m)}, \mathbf{Q}_0^{(m)}\right)$:

$$p\left(\mathbf{x}_0^{(m)}\right)\prod_{t=1}^{T}p\left(\mathbf{x}_t^{(m)}|\mathbf{x}_{t-1}^{(m)}\right) =$$

$$\left|2\pi\mathbf{Q}_0^{(m)}\right|^{-1/2}\exp\left\{ -\frac{1}{2}\left(\mathbf{x}_0^{(m)} - \boldsymbol{\mu}^{(m)}\right)^{\top}\mathbf{Q}_0^{(m)-1}\left(\mathbf{x}_0^{(m)} - \boldsymbol{\mu}^{(m)}\right)\right\} \times \tag{8}$$

$$\prod_{t=1}^{T}\left|2\pi\mathbf{Q}^{(m)}\right|^{-1/2}\exp\left\{ -\frac{1}{2}\left(\mathbf{x}_t^{(m)} - \mathbf{F}^{(m)}\mathbf{x}_{t-1}^{(m)}\right)^{\top}\mathbf{Q}^{(m)-1}\left(\mathbf{x}_t^{(m)} - \mathbf{F}^{(m)}\mathbf{x}_{t-1}^{(m)}\right)\right\}$$

Lastly, the discrete-valued switching state evolves according to the HMM specified by the $M \times M$ state-transition matrix $\boldsymbol{\phi}$ and initial state probabilities $\boldsymbol{\rho}$, independent of the other $M$

Gaussian SSMs with real-valued states:

$$p(s_0)\prod_{t=1}^{T}p(s_t|s_{t-1}) = \prod_{m=1}^{M}\boldsymbol{\rho}_m^{\mathbb{1}[s_0=m]}\prod_{t=1}^{T}\prod_{m,n=1}^{M}\boldsymbol{\phi}_{m,n}^{\mathbb{1}[s_t=m,s_{t-1}=n]} \tag{9}$$

**Intractable posterior of hidden states.** With the generative model defined, state estimation and system identification problems need to be solved. As reviewed above, both problems are encompassed in the EM algorithm, which alternates between 1) finding the posterior distribution of the hidden states $\left\{\mathbf{x}_t^{(1)}, \cdots, \mathbf{x}_t^{(M)}, s_t\right\}$ given the current values of model parameters, and 2) optimizing the model parameters given expectations of hidden states under the posterior distribution from (1). However, the hidden state posterior here cannot be computed explicitly. To illustrate this difficulty, we start with the complete log-likelihood of observations and hidden states:

$$\log p\left(\left\{\mathbf{y}_t, \mathbf{x}_t^{(1)}, \cdots, \mathbf{x}_t^{(M)}, s_t\right\}|\boldsymbol{\theta}\right)$$
$$= -\frac{T}{2}\log|2\pi\mathbf{R}| - \frac{1}{2}\sum_{t=1}^{T}\sum_{m=1}^{M}\mathbb{1}[s_t=m]\left(\mathbf{y}_t - \mathbf{G}^{(m)}\mathbf{x}_t^{(m)}\right)^{\top}\mathbf{R}^{-1}\left(\mathbf{y}_t - \mathbf{G}^{(m)}\mathbf{x}_t^{(m)}\right)$$
$$-\frac{1}{2}\sum_{m=1}^{M}\log\left|2\pi\mathbf{Q}_0^{(m)}\right| - \frac{1}{2}\sum_{m=1}^{M}\left(\mathbf{x}_0^{(m)} - \boldsymbol{\mu}^{(m)}\right)^{\top}\mathbf{Q}_0^{(m)-1}\left(\mathbf{x}_0^{(m)} - \boldsymbol{\mu}^{(m)}\right) \tag{10}$$
$$-\frac{T}{2}\sum_{m=1}^{M}\log\left|2\pi\mathbf{Q}^{(m)}\right| - \frac{1}{2}\sum_{m=1}^{M}\sum_{t=1}^{T}\left(\mathbf{x}_t^{(m)} - \mathbf{F}^{(m)}\mathbf{x}_{t-1}^{(m)}\right)^{\top}\mathbf{Q}^{(m)-1}\left(\mathbf{x}_t^{(m)} - \mathbf{F}^{(m)}\mathbf{x}_{t-1}^{(m)}\right)$$
$$+\sum_{m=1}^{M}\mathbb{1}[s_0=m]\log\boldsymbol{\rho}_m + \sum_{t=1}^{T}\sum_{m,n=1}^{M}\mathbb{1}[s_t=m,s_{t-1}=n]\log\boldsymbol{\phi}_{m,n}$$

where $\boldsymbol{\theta} \triangleq \left\{\left\{\boldsymbol{\mu}^{(m)}, \mathbf{Q}_0^{(m)}, \mathbf{F}^{(m)}, \mathbf{Q}^{(m)}, \mathbf{G}^{(m)}\right\}, \boldsymbol{\rho}, \boldsymbol{\phi}, \mathbf{R}\right\}$, for $m = 1, \cdots, M$, is the set of parameters of the distributions that fully characterize the switching state-space generative model in Fig 9. Note that the HMM process does not have a separate observation probability parameter (e.g. $\boldsymbol{\psi}$). Once the switching state is fixed to $s_t = m$, the observation equation dynamic of the corresponding $m^{\text{th}}$ Gaussian SSM in the second term of Eq 10 specifies the likelihood of generating the observation at that time point.

State estimation, i.e., the E-step, is computationally difficult because of the products between the discrete switching states of the HMM and the real-valued states of the Gaussian SSMs. This conditional dependence between the hidden states (the 'two' causes) when conditioning on the observations (the 'common' effect) has been described in the causal inference literature [102, pp.40–45]. Graphically, conditioning on the children of v-structures results in a graph with parents connected with undirected edges for the new conditional (in)dependence relation (Fig 10a). The new edges introduced make it challenging to express the exact hidden state posterior in a factorized form. In particular, the individual Markov independence property within each Gaussian SSM is no longer present, since conditioning on $\mathbf{x}_{t-1}^{(m)}|\{\mathbf{y}_t\}$ alone does not make $\mathbf{x}_t^{(m)}|\{\mathbf{y}_t\}$ conditionally independent from the history $\mathbf{x}_{0,\cdots,t-2}^{(m)}|\{\mathbf{y}_t\}$ due to the new viable paths through the other Gaussian SSMs. Then exact inferences can only be solved after considering the combinations of states across all Gaussian SSMs and HMM. For example, the optimal filtering problem alone requires a bank of elemental estimators with the size of the bank increasing exponentially with time [32], so naturally the smoothing problem, i.e., state

estimation of $p\left(\left\{\mathbf{x}_t^{(1)}, \cdots, \mathbf{x}_t^{(M)}, s_t\right\} | \{\mathbf{y}_t\}\right)$, becomes intractable. Without the hidden state posterior distribution estimated, system identification, i.e., parameter learning, cannot be performed either under the classical EM algorithm [93].

**Variational approximation of hidden state posterior.** One possible solution is to use the Variational Bayes technique to approximate the true posterior of the hidden states, $p\left(\left\{\mathbf{x}_t^{(1)}, \cdots, \mathbf{x}_t^{(M)}, s_t\right\} | \{\mathbf{y}_t\}\right)$ [53]. The idea is to work within a subspace of tractable posterior probability distributions defined over the hidden states, and choose the optimal approximation, $q\left(\left\{\mathbf{x}_t^{(1)}, \cdots, \mathbf{x}_t^{(M)}, s_t\right\}\right)$, based on a lower bound on the marginal log-likelihood of observations:

$$
\begin{aligned}
\log p(\{\mathbf{y}_t\}|\boldsymbol{\theta}) \quad &= \log \sum_{\{s_t\}} \int d\{\mathbf{x}_t\} p\left(\left\{\mathbf{y}_t, \mathbf{x}_t^{(1)}, \cdots, \mathbf{x}_t^{(M)}, s_t\right\}|\boldsymbol{\theta}\right) \\
&= \log \sum_{\{s_t\}} \int d\left\{\mathbf{x}_t^{(1)}, \cdots, \mathbf{x}_t^{(M)}\right\} q\left(\left\{\mathbf{x}_t^{(1)}, \cdots, \mathbf{x}_t^{(M)}, s_t\right\}\right) \times \\
&\qquad\qquad \frac{p\left(\left\{\mathbf{y}_t, \mathbf{x}_t^{(1)}, \cdots, \mathbf{x}_t^{(M)}, s_t\right\}|\boldsymbol{\theta}\right)}{q\left(\left\{\mathbf{x}_t^{(1)}, \cdots, \mathbf{x}_t^{(M)}, s_t\right\}\right)} \\
&\geq \sum_{\{s_t\}} \int d\left\{\mathbf{x}_t^{(1)}, \cdots, \mathbf{x}_t^{(M)}\right\} q\left(\left\{\mathbf{x}_t^{(1)}, \cdots, \mathbf{x}_t^{(M)}, s_t\right\}\right) \times \\
&\qquad\qquad \log \frac{p\left(\left\{\mathbf{y}_t, \mathbf{x}_t^{(1)}, \cdots, \mathbf{x}_t^{(M)}, s_t\right\}|\boldsymbol{\theta}\right)}{q\left(\left\{\mathbf{x}_t^{(1)}, \cdots, \mathbf{x}_t^{(M)}, s_t\right\}\right)} \\
&= \mathbb{E}_q \left[ \log \frac{p\left(\left\{\mathbf{y}_t, \mathbf{x}_t^{(1)}, \cdots, \mathbf{x}_t^{(M)}, s_t\right\}|\boldsymbol{\theta}\right)}{q\left(\left\{\mathbf{x}_t^{(1)}, \cdots, \mathbf{x}_t^{(M)}, s_t\right\}\right)} \right] \triangleq \mathcal{F}(q, \boldsymbol{\theta})
\end{aligned}
\tag{11}
$$

known as the *negative variational free energy* [54, 103, 104]. Since the distribution $q$ is approximating the true posterior, the conditioning on $\{\mathbf{y}_t\}$ in the expression $q\left(\left\{\mathbf{x}_t^{(1)}, \cdots, \mathbf{x}_t^{(M)}, s_t \mid \{\mathbf{y}_t\}\right\}\right)$ is already implied, therefore omitted in all $q$ notations.

The choice of $q\left(\left\{\mathbf{x}_t^{(1)}, \cdots, \mathbf{x}_t^{(M)}, s_t\right\}\right) = p\left(\left\{\mathbf{x}_t^{(1)}, \cdots, \mathbf{x}_t^{(M)}, s_t\right\}|\{\mathbf{y}_t\}\right)$ maximizes the negative free energy so that it reaches the true log-likelihood. However, since exact E-step is intractable due to the structure of the posterior distribution, we consider the following family of distributions that factor over the HMM and Gaussian SSMs, i.e.,

$$
\mathcal{P} = \left\{ q\left(\left\{\mathbf{x}_t^{(1)}, \cdots, \mathbf{x}_t^{(M)}, s_t\right\}\right) \;\middle|\; q(\bullet) = q(\{s_t\}) q\left(\left\{\mathbf{x}_t^{(1)}, \cdots, \mathbf{x}_t^{(M)}\right\}\right) \right\}
\tag{12}
$$

In other words, we get rid of the dependence between the switching state and real-valued states by limiting to distributions within this functional subspace. The graphical model corresponding to this conditional independence relation is shown in Fig 10b. It only remains to find the optimal approximate posterior $q\left(\left\{\mathbf{x}_t^{(1)}, \cdots, \mathbf{x}_t^{(M)}, s_t\right\}\right)$ that is the *closest* to the true posterior $p\left(\left\{\mathbf{x}_t^{(1)}, \cdots, \mathbf{x}_t^{(M)}, s_t\right\}|\{\mathbf{y}_t\}\right)$. Since negative free energy is a lower bound on the marginal log-likelihood, we use the negative free energy as a measure of one-sided *closeness* to

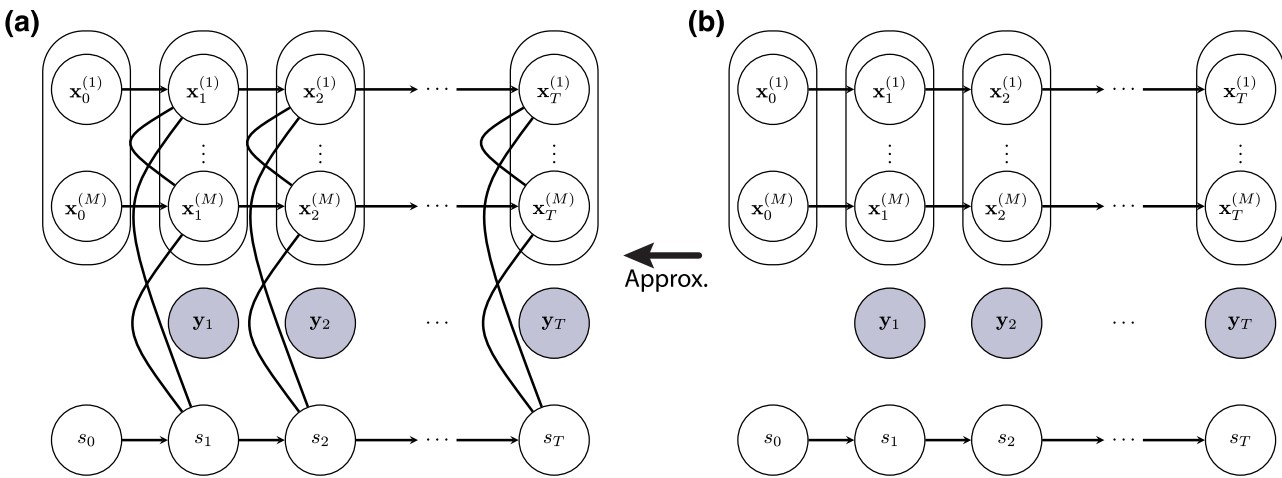

**Fig 10. Posteriors of switching state-space models.** (a) True posterior distribution. This is the resultant graph encoding the conditional independence relation after conditioning on the observed data $\{\mathbf{y}_t\}$ up to time $T$. The new edges between the hidden states make exact inference intractable. (b) Approximate posterior distribution. Compared to the true posterior, a structured approximation decouples the hidden Gaussian state-space models from each other and from the switching state. On this approximate distribution, efficient closed-form inference can be performed. The marginal distributions of the Gaussian hidden states $\{\mathbf{x}_t\}$ and the discrete-valued switching state $\{s_t\}$ are now inter-dependent through variational summary statistics $g_t^{(m)}$ and $h_t^{(m)}$.

choose the optimal approximation as following:

$$q_{\boldsymbol{\theta}} = \arg\max_{q\in\mathcal{P}} \mathcal{F}(q, \boldsymbol{\theta}) \tag{13}$$

We proceed to a direct derivation of the optimal functional forms of the approximate posterior. Within the functional subspace $\mathcal{P}$, the negative free energy can be simplified:

$$
\begin{aligned}
\mathcal{F}(q, \boldsymbol{\theta}) \quad &= \left\langle \left\langle \log p\left(\left\{\mathbf{y}_t, \mathbf{x}_t^{(1)}, \cdots, \mathbf{x}_t^{(M)}, s_t\right\}|\boldsymbol{\theta}\right) \right\rangle_s \right\rangle_x \\
&\quad - \left\langle \log q(\{s_t\})\right\rangle_s - \left\langle \log q\left(\left\{\mathbf{x}_t^{(1)}, \cdots, \mathbf{x}_t^{(M)}\right\}\right) \right\rangle_x
\end{aligned}
\tag{14}
$$

where

$$\langle f \rangle_s \triangleq \sum_{\{s_t\}} q(\{s_t\}) f$$

$$\langle f \rangle_x \triangleq \int d\left\{\mathbf{x}_t^{(1)}, \cdots, \mathbf{x}_t^{(M)}\right\} q\left(\left\{\mathbf{x}_t^{(1)}, \cdots, \mathbf{x}_t^{(M)}\right\}\right) f$$

Following the development in [54], the optimal functional forms for the variational log-posteriors to maximize Eq 14 are given as:

$$\log q(\{s_t\}) = \left\langle \log p\left(\left\{\mathbf{y}_t, \mathbf{x}_t^{(1)}, \cdots, \mathbf{x}_t^{(M)}, s_t\right\}|\boldsymbol{\theta}\right) \right\rangle_x - \log z \tag{15}$$

$$\log q\left(\left\{\mathbf{x}_t^{(1)}, \cdots, \mathbf{x}_t^{(M)}\right\}\right) = \left\langle \log p\left(\left\{\mathbf{y}_t, \mathbf{x}_t^{(1)}, \cdots, \mathbf{x}_t^{(M)}, s_t\right\}|\boldsymbol{\theta}\right) \right\rangle_s - \log z' \tag{16}$$

with $z$ and $z'$ being normalizing constants so that the posteriors sum or integrate to 1.

Substituting the expressions from Eq 10 into Eq 15 reveals that the variational log-posterior of the switching state can be written as:

$$
\log q(\{s_t\}) = \sum_{m=1}^{M} \mathbb{1}[s_0 = m]\log \boldsymbol{\rho}_m + \sum_{t=1}^{T}\sum_{m,n=1}^{M} \mathbb{1}[s_t = m, s_{t-1} = n] \log \boldsymbol{\phi}_{m,n}
$$
$$
+ \sum_{t=1}^{T}\sum_{m=1}^{M} \mathbb{1}[s_t = m]g_t^{(m)} - \log \zeta \tag{17}
$$

$$
\text{with } g_t^{(m)} \triangleq -\frac{1}{2}\left\langle \left(\mathbf{y}_t - \mathbf{G}^{(m)}\mathbf{x}_t^{(m)}\right)^{\top} \mathbf{R}^{-1}\left(\mathbf{y}_t - \mathbf{G}^{(m)}\mathbf{x}_t^{(m)}\right)\right\rangle_x \tag{18}
$$

This equation is identical to the log-posterior density of the discrete-valued states in an HMM with observation probabilities proportional to $\exp\{g_t^{(m)}\}$. In other words, $g_t^{(m)}$ functions as the model log-evidence of the $m^{\text{th}}$ state-space model in generating the data at time $t$ in the HMM process. Thus, the desired posterior distribution, i.e., $q(s_t = m)$ at individual time points, can be expressed in terms of the forward-backward variables $(\alpha_t(m), \beta_t(m))$ from the forward-backward algorithm [94] and computed efficiently (S3 Appendix).

Similarly, using the expressions from Eqs 10 and 16, we obtain the following variational log-posterior of the real-valued states in the Gaussian SSMs:

$$
\log q\left(\left\{\mathbf{x}_t^{(1)},\cdots,\mathbf{x}_t^{(M)}\right\}\right) = -\frac{1}{2}\sum_{m=1}^{M}\left(\mathbf{x}_0^{(m)} - \boldsymbol{\mu}^{(m)}\right)^{\top} \mathbf{Q}_0^{(m)-1}\left(\mathbf{x}_0^{(m)} - \boldsymbol{\mu}^{(m)}\right)
$$
$$
-\frac{1}{2}\sum_{m=1}^{M}\sum_{t=1}^{T}\left(\mathbf{x}_t^{(m)} - \mathbf{F}^{(m)}\mathbf{x}_{t-1}^{(m)}\right)^{\top} \mathbf{Q}^{(m)-1}\left(\mathbf{x}_t^{(m)} - \mathbf{F}^{(m)}\mathbf{x}_{t-1}^{(m)}\right) \tag{19}
$$
$$
-\frac{1}{2}\sum_{t=1}^{T}\sum_{m=1}^{M} h_t^{(m)}\left(\mathbf{y}_t - \mathbf{G}^{(m)}\mathbf{x}_t^{(m)}\right)^{\top} \mathbf{R}^{-1}\left(\mathbf{y}_t - \mathbf{G}^{(m)}\mathbf{x}_t^{(m)}\right) - \log \zeta'
$$

$$
\text{with } h_t^{(m)} \triangleq \langle \mathbb{1}[s_t = m]\rangle_s \tag{20}
$$

that factorizes over the $M$ parallel state-space models, i.e.,

$$
\log q\left(\left\{\mathbf{x}_t^{(1)},\cdots,\mathbf{x}_t^{(M)}\right\}\right) = \sum_{m=1}^{M} \log q\left(\left\{\mathbf{x}_t^{(m)}\right\}\right) \tag{21}
$$

Each of the log-posteriors within the sum takes the familiar Gaussian density form:

$$
\log q\quad\left(\left\{\mathbf{x}_t^{(m)}\right\}\right) = -\frac{1}{2}\left(\mathbf{x}_0^{(m)} - \boldsymbol{\mu}^{(m)}\right)^{\top} \mathbf{Q}_0^{(m)-1}\left(\mathbf{x}_0^{(m)} - \boldsymbol{\mu}^{(m)}\right)
$$
$$
-\frac{1}{2}\sum_{t=1}^{T}\left(\mathbf{x}_t^{(m)} - \mathbf{F}^{(m)}\mathbf{x}_{t-1}^{(m)}\right)^{\top} \mathbf{Q}^{(m)-1}\left(\mathbf{x}_t^{(m)} - \mathbf{F}^{(m)}\mathbf{x}_{t-1}^{(m)}\right) \tag{22}
$$
$$
-\frac{1}{2}\sum_{t=1}^{T} h_t^{(m)}\left(\mathbf{y}_t - \mathbf{G}^{(m)}\mathbf{x}_t^{(m)}\right)^{\top} \mathbf{R}^{-1}\left(\mathbf{y}_t - \mathbf{G}^{(m)}\mathbf{x}_t^{(m)}\right) - \log \zeta'_m
$$

that can be computed using a forward pass through Kalman filter, followed by a backward pass through RTS smoother to get the posterior mean and covariance matrices, $\left\{\mathbf{x}_{t|T}^{(m)}, \boldsymbol{\Sigma}_{t|T}^{(m)}, \boldsymbol{\Sigma}_{t,t-1|T}^{(m)}\right\}$, as shown in S3 Appendix.

Given the third term in Eq 22, Kalman smoothing here should be modified to use $\mathbf{R}/h_t^{(m)}$ as the observation noise covariance. Scaling the observation noise leads to an intuitive interpretation: when $h_t^{(m)} \to 0$, resulting in high effective observation noise, the hidden state update that follows one-step prediction during Kalman filtering is effectively skipped for that time point; when $h_t^{(m)} \to 1$, the hidden state is updated as in recursions without switching. In this sense, $h_t^{(m)}$ represents the model responsibility of the $m^{\text{th}}$ state-space model in generating the observed data at the time point $t$.

**Fixed-point iterations.** The variables $g_t^{(m)}$ and $h_t^{(m)}$ are the bridges between the distributions $p$ and $q$, replacing the stochastic dependencies with deterministic quantities. The variational log-posterior of the discrete switching states $\{s_t\}$ in Eq 18 involves some summary statistics of the real-valued states $\{\mathbf{x}_t^{(1)}, \cdots, \mathbf{x}_t^{(M)}\}$ w.r.t. the approximate distribution $q$, and vice-versa in Eq 20. This motivates an iterative procedure to update the forward-backward variables and posterior means and covariances in a cyclic fashion to find the optimal variational parameters $\left\{ \alpha_t(m), \beta_t(m), \mathbf{x}_{t|T}^{(m)}, \mathbf{\Sigma}_{t|T}^{(m)}, \mathbf{\Sigma}_{t,t-1|T}^{(m)} \right\}$:

1. Run forward-backward algorithm to update $\{\alpha_t(m), \beta_t(m)\}$.

2. Compute $h_t^{(m)}$ from the updated $\{\alpha_t(m), \beta_t(m)\}$.

3. Run Kalman filter and RTS smoother to update $\left\{ \mathbf{x}_{t|T}^{(m)}, \mathbf{\Sigma}_{t|T}^{(m)}, \mathbf{\Sigma}_{t,t-1|T}^{(m)} \right\}$.

4. Compute $g_t^{(m)}$ from the updated $\left\{ \mathbf{x}_{t|T}^{(m)}, \mathbf{\Sigma}_{t|T}^{(m)} \right\}$.

During these iterations, the negative variational free energy can be computed using log-likelihood values from the Kalman filter and forward-backward algorithm as shown in S1 Appendix. We stop the fixed-point iterations when the model evidence and model responsibility, i.e., $\{g_t^{(m)}, h_t^{(m)}\}$, have stabilized or when the negative free energy stops increasing (see S3 Appendix).

Under the assumed generative structure in Fig 9, the true posterior $p$ is expected to contain polarized values (e.g., with model probabilities close to 1 or 0) due to a few factors. First, each transition between candidate models introduces a *discontinuity* in the observation sequence at the switch point, as the realization trajectories from different SSMs are distinct from each other. Second, since the SSM hidden state trajectories consist of multiple time points, they reside in a high dimensional space. The dimensionality further increases as SSM state dimension increases. Because of this high-dimensionality, the separation between any two candidate models becomes very substantial. Third, real recordings often have infrequent transitions between distinct dynamics. This empirical skewness leads to a high probability to stay within the same candidate model. Such probability propagates as a strong prior across time points amplifying the effect of the other factors in polarizing the values of posterior estimates.

Correspondingly, the fixed-point iterations could polarize the values of $h_t^{(m)}$ through the reciprocal interdependence between $g_t^{(m)}$ and $h_t^{(m)}$. The observation noise covariance during Kalman smoothing is scaled by the model responsibility as $\mathbf{R}/h_t^{(m)}$ to produce the model log-evidence following Eq 18. This log-evidence is then used by the forward-backward algorithm to compute the model responsibility for the next pass of fixed-point iterations. Thus, this cycle could amplify the $h_t^{(m)}$ for the best candidate model toward 1 while pushing the others close to 0, effectively assigning each time point to one of the candidate models instead of maintaining a mixture of models. This empirical behavior of the fixed-point iterations appears similar to

automatic relevance determination [105], with $\mathbf{G}^{(m)}\mathbf{x}_t^{(m)}$ treated as features weighted by $h_t^{(m)}$ at each time point. However, the log-likelihood expression and therefore the pruning mechanism of the respective solutions are different.

It should be emphasized that the negative variational free energy is not jointly concave w.r. t. the values of $g_t^{(m)}$ and $h_t^{(m)}$. In fact, the free energy landscape contains abundant local maxima, and the fixed-point iterations stop updating whenever a local maximum is reached. Empirically, the fixed-point iterations indeed recover polarized $h_t^{(m)}$ even when the segmentation accuracy is low, i.e., when sub-optimal local maxima are reached. This behavior makes proper initialization the most important factor in the performance of the current variational inference approach. We address this technical challenge below in section Initialization of fixed-point iterations.

**Parameter learning using generalized EM algorithm.** Once the approximation of the hidden state posterior can be computed efficiently, one can employ an EM-like algorithm to learn the model parameters of the Gaussian SSMs and HMM. To this end, starting from an initial guess of model parameters, we first repeat the fixed-point iterations until convergence to find the optimal variational parameters, $\left\{ \alpha_t(m), \beta_t(m), \mathbf{x}_{t|T}^{(m)}, \mathbf{\Sigma}_{t|T}^{(m)}, \mathbf{\Sigma}_{t,t-1|T}^{(m)} \right\}$, which parameterize the approximate hidden state posterior (E-step). Next, we update the model parameters of the individual Gaussian SSMs and HMM to maximize the negative variational free energy using the converged variational parameters (M-step). Iterating between these two steps constitutes an instance of *generalized* EM algorithm where in lieu of data log-likelihood, a lower bound on the log-likelihood is maximized over successive iterations [106]. It should be noted that while derivations differ, the overall form of this algorithm is the same as in [53]. A flowchart of this variational learning algorithm is shown in Fig 1.

In the following section, we derive closed-form update equations for the model parameters of the Gaussian SSMs and HMM, given the approximate hidden state posterior. We note that these M-step equations can be derived from the complete log-likelihood expression in Eq 10 under the expectation w.r.t. the variational approximate distribution $q$, because only the first term in Eq 14 depends on the parameters $\boldsymbol{\theta}$. It is therefore sufficient to maximize this term for an optimization problem analogous to Eq 13 but over $\boldsymbol{\theta}$.

Specifically, at the $i^{\text{th}}$ EM iteration, with the converged variational parameters, $\left\{ \alpha_t(m), \beta_t(m), \mathbf{x}_{t|T}^{(m)}, \mathbf{\Sigma}_{t|T}^{(m)}, \mathbf{\Sigma}_{t,t-1|T}^{(m)} \right\}^i$, we compute the following summary statistics:

$$\mathbf{A}^{(m)} \triangleq \sum_{t=1}^{T} \mathbb{E}\left[ \mathbf{x}_{t-1}^{(m)}\mathbf{x}_{t-1}^{(m)\top} \right] = \sum_{t=1}^{T} \mathbf{\Sigma}_{t-1|T}^{(m)} + \mathbf{x}_{t-1|T}^{(m)}\mathbf{x}_{t-1|T}^{(m)\top}$$

$$\mathbf{B}^{(m)} \triangleq \sum_{t=1}^{T} \mathbb{E}\left[ \mathbf{x}_{t}^{(m)}\mathbf{x}_{t-1}^{(m)\top} \right] = \sum_{t=1}^{T} \mathbf{\Sigma}_{t,t-1|T}^{(m)} + \mathbf{x}_{t|T}^{(m)}\mathbf{x}_{t-1|T}^{(m)\top}$$

$$\mathbf{C}^{(m)} \triangleq \sum_{t=1}^{T} \mathbb{E}\left[ \mathbf{x}_{t}^{(m)}\mathbf{x}_{t}^{(m)\top} \right] = \sum_{t=1}^{T} \mathbf{\Sigma}_{t|T}^{(m)} + \mathbf{x}_{t|T}^{(m)}\mathbf{x}_{t|T}^{(m)\top}$$

$$p_{t|T}(m) \triangleq \mathbb{P}[s_t = m] = \mathbb{E}[\mathbb{1}[s_t = m]] = h_t^{(m)}$$

$$p_{t,t-1|T}(m,n) \triangleq \mathbb{P}[s_t = m, s_{t-1} = n] = \mathbb{E}[\mathbb{1}[s_t = m, s_{t-1} = n]]$$

S3 Appendix provides full recursions used to compute these summary statistics, which are

needed to take the expectation of the complete log-likelihood in Eq 10:

$$
\mathbb{E}\left[\log p\left(\left\{\mathbf{y}_t, \mathbf{x}_t^{(1)}, \cdots, \mathbf{x}_t^{(M)}, s_t\right\} | \boldsymbol{\theta}\right)\right]
$$

$$
= -\frac{T}{2}\log|2\pi\mathbf{R}| - \frac{1}{2}\sum_{m=1}^{M}\log\left|2\pi\mathbf{Q}_0^{(m)}\right| - \frac{T}{2}\sum_{m=1}^{M}\log|2\pi\mathbf{Q}^{(m)}|
$$

$$
- \frac{1}{2}\sum_{m=1}^{M}\sum_{t=1}^{T}h_t^{(m)}\mathrm{Tr}\left\{\mathbf{R}^{-1}\left(\left(\mathbf{y}_t - \mathbf{G}^{(m)}\mathbf{x}_{t|T}^{(m)}\right)\left(\mathbf{y}_t - \mathbf{G}^{(m)}\mathbf{x}_{t|T}^{(m)}\right)^{\top} + \mathbf{G}^{(m)}\boldsymbol{\Sigma}_{t|T}^{(m)}\mathbf{G}^{(m)\top}\right)\right\}
$$

$$
- \frac{1}{2}\sum_{m=1}^{M}\mathrm{Tr}\left\{\mathbf{Q}_0^{(m)^{-1}}\left(\left(\mathbf{x}_{0|T}^{(m)} - \boldsymbol{\mu}^{(m)}\right)\left(\mathbf{x}_{0|T}^{(m)} - \boldsymbol{\mu}^{(m)}\right)^{\top} + \boldsymbol{\Sigma}_{0|T}^{(m)}\right)\right\} \tag{23}
$$

$$
- \frac{1}{2}\sum_{m=1}^{M}\mathrm{Tr}\left\{\mathbf{Q}^{(m)^{-1}}\left(\mathbf{C}^{(m)} - \mathbf{B}^{(m)}\mathbf{F}^{(m)\top} - \mathbf{F}^{(m)}\mathbf{B}^{(m)\top} + \mathbf{F}^{(m)}\mathbf{A}^{(m)}\mathbf{F}^{(m)\top}\right)\right\}
$$

$$
+ \sum_{m=1}^{M}p_{0|T}(m)\log\boldsymbol{\rho}_m + \sum_{t=1}^{T}\sum_{m,n=1}^{M}p_{t,t-1|T}(m, n)\log\boldsymbol{\phi}_{m,n}
$$

The optimal parameters, $\boldsymbol{\theta}^i \triangleq \left\{\left\{\boldsymbol{\mu}^{(m)i}, \mathbf{Q}_0^{(m)i}, \mathbf{F}^{(m)i}, \mathbf{Q}^{(m)i}, \mathbf{G}^{(m)i}\right\}, \boldsymbol{\rho}^i, \boldsymbol{\phi}^i, \mathbf{R}^i\right\}$ for $m = 1, \cdots, M$, which maximize Eq 23, can be obtained in closed-form by taking individual partial derivatives. The parameter subset $\left\{\boldsymbol{\mu}^{(m)}, \mathbf{Q}_0^{(m)}, \mathbf{F}^{(m)}, \mathbf{Q}^{(m)}, \mathbf{G}^{(m)}\right\}$ can be updated for each Gaussian SSM through the usual equations described in [93]:

$$
\boldsymbol{\mu}^{(m)i} = \mathbf{x}_{0|T}^{(m)} \tag{24}
$$

$$
\mathbf{Q}_0^{(m)i} = \boldsymbol{\Sigma}_{0|T}^{(m)} + \mathbf{x}_{0|T}^{(m)}\mathbf{x}_{0|T}^{(m)\top} - \mathbf{x}_{0|T}^{(m)}\boldsymbol{\mu}^{(m)i\top} - \boldsymbol{\mu}^{(m)i}\mathbf{x}_{0|T}^{(m)\top} + \boldsymbol{\mu}^{(m)i}\boldsymbol{\mu}^{(m)i\top} \tag{25}
$$

$$
\mathbf{F}^{(m)i} = \mathbf{B}^{(m)}\mathbf{A}^{(m)^{-1}} \tag{26}
$$

$$
\mathbf{Q}^{(m)i} = \frac{1}{T}\left(\mathbf{C}^{(m)} - \mathbf{B}^{(m)}\mathbf{F}^{(m)i\top} - \mathbf{F}^{(m)i}\mathbf{B}^{(m)\top} + \mathbf{F}^{(m)i}\mathbf{A}^{(m)}\mathbf{F}^{(m)i\top}\right) \tag{27}
$$

with a slight exception for $\mathbf{G}^{(m)}$ due to the product with the switching state:

$$
\mathbf{G}^{(m)i} = \left(\sum_{t=1}^{T}h_t^{(m)}\mathbf{y}_t\mathbf{x}_{t|T}^{(m)\top}\right)\left(\sum_{t=1}^{T}h_t^{(m)}\left(\boldsymbol{\Sigma}_{t|T}^{(m)} + \mathbf{x}_{t|T}^{(m)}\mathbf{x}_{t|T}^{(m)\top}\right)\right)^{-1} \tag{28}
$$

We note that since the $M$ Gaussian SSMs have independent dynamics, these updates can be completed efficiently in parallel.

Similarly, the usual update equations for an HMM can be used to update $\{\boldsymbol{\rho}, \boldsymbol{\phi}\}$ [101]:

$$
\boldsymbol{\rho}_m^i = p_{0|T}(m) \tag{29}
$$

$$
\boldsymbol{\phi}_{m,n}^i = \frac{\sum_{t=1}^{T}p_{t,t-1|T}(m, n)}{\sum_{t=1}^{T}\sum_{m=1}^{M}p_{t,t-1|T}(m, n)} \tag{30}
$$

As noted earlier, the observation probabilities of the HMM are not explicitly updated as a parameter, since the variational model evidence, $g_t^{(m)}$, is converged through the fixed-point

iterations and used as point estimates of the (log) observation probability to relate the hidden states of Gaussian SSMs to the observation at each time point.

Finally, the update equation for $\mathbf{R}$ pools the posterior estimates of hidden states across the $M$ Gaussian SSMs:

$$\mathbf{R}^i = \frac{1}{T}\sum_{m=1}^{M}\sum_{t=1}^{T} h_t^{(m)}\mathbf{\Omega}_t^{(m)} \tag{31}$$

where $\mathbf{\Omega}_t^{(m)}$ captures the contribution from the $m^{\text{th}}$ model:

$$\mathbf{\Omega}_t^{(m)} = \left(\mathbf{y}_t - \mathbf{G}^{(m)i}\mathbf{x}_{t|T}^{(m)}\right)\left(\mathbf{y}_t - \mathbf{G}^{(m)i}\mathbf{x}_{t|T}^{(m)}\right)^{\top} + \mathbf{G}^{(m)i}\mathbf{\Sigma}_{t|T}^{(m)}\mathbf{G}^{(m)i\top} \tag{32}$$

This update equation for $\mathbf{R}$ is an instance of joint estimation of parameters shared among Gaussian SSMs. In a more general case where each state-space model has its individual observation noise covariance $\mathbf{R}^{(m)}$, the update equation takes the form:

$$\mathbf{R}^{(m)i} = \frac{1}{\sum_{t=1}^{T} h_t^{(m)}}\sum_{t=1}^{T} h_t^{(m)}\mathbf{\Omega}_t^{(m)} \tag{33}$$

It is also possible for other Gaussian SSM parameters $\left\{\boldsymbol{\mu}^{(m)}, \mathbf{Q}_0^{(m)}, \mathbf{F}^{(m)}, \mathbf{Q}^{(m)}, \mathbf{G}^{(m)}\right\}$ to be partially shared among different models. Closed-form update equations in these cases can be derived. This extension exploits the flexibility of the variational Bayesian method to accommodate different generative models. Two such examples are studied in this paper, and we provide the derivations of their update equations in S2 Appendix.

**Initialization of fixed-point iterations.**   As shown above, the M-step equations provide optimal updates of parameters in a convex optimization setting given the multivariate Gaussian distribution $q$ and the posterior hidden state estimates. In contrast, the fixed-point iterations in the E-step are not guaranteed to achieve globally maximal negative free energy due to the non-concavity of the negative free energy w.r.t. the variational parameters. This makes the final approximate distribution at the end of fixed-point iterations very sensitive to initialization. Moreover, while the fixed-point iterations are defined explicitly step-by-step, it is not obvious how to initialize these iterations. Thus, here we describe two practical initialization approaches.

**Deterministic annealing.** This entropy-motivated initialization technique was proposed along with the variational framework to alleviate getting trapped in local maxima during the fixed-point iterations [53, 107]. Specifically, equal model responsibilities are used as the initial $h_t^{(m)}$ at the onset of the first E-step, followed by RTS smoothing and forward-backward algorithm to compute $g_t^{(m)}$ and $h_t^{(m)}$ with definitions modified by a temperature parameter $\mathcal{T}$:

$$g_t^{(m)} = \frac{1}{\mathcal{T}}\left(-\frac{1}{2}\left\langle\left(\mathbf{y}_t - \mathbf{G}^{(m)}\mathbf{x}_t^{(m)}\right)^{\top}\mathbf{R}^{-1}\left(\mathbf{y}_t - \mathbf{G}^{(m)}\mathbf{x}_t^{(m)}\right)\right\rangle_x\right) \tag{34}$$

$$h_t^{(m)} = \frac{1}{\mathcal{T}}\left(\langle\mathbb{1}[s_t = m]\rangle_s\right) \tag{35}$$

The temperature $\mathcal{T}$ cools over successive fixed-point iterations via the decay function $\mathcal{T}_{i+1} = (\mathcal{T}_i + 1)/2$. In essence, a larger $\mathcal{T}$ allows the iterations to explore a bigger subspace with high negative free energy entropy, which gradually decreases in trying to identify an

optimal approximate distribution. This initialization technique was necessary for the variational approximation to produce any reasonable results [53].

After the first E-step, $h_t^{(m)}$ is not re-initialized with equal model responsibilities to allow a warm start in subsequent iterations. However, this makes the algorithm severely limited by the results of the first round of fixed-point iterations. Under this initialization scheme, once the $h_t^{(m)}$ for the $m^{\text{th}}$ model gets close to zero, that model cannot regain responsibility for that time point in subsequent iterations. One may choose to reset the $h_t^{(m)}$ to be equal across models at every E-step, but that tends to select only one of the Gaussian SSMs for the entire duration as a local maximum. Regardless, with the model parameters fixed during the E-step, potential local maxima are likely selected, since the annealing is equivalent across the $M$ Gaussian SSMs. Also, initializing $h_t^{(m)}$ with equal responsibilities assumes all models to explain all time points equally well, which is certainly far from the true posterior $p$. Thus, a better initialization should discern among the Gaussian SSMs and try to initialize closer to the global maximum.

**Interpolated densities.** Here we propose a different initialization technique that statistically compares the $M$ Gaussian SSMs for their likelihoods of generating the observation at each time point. Specifically, we initialize $g_t^{(m)}$ using the interpolated density [108, 109], i.e., the probability density of the observation at a given time point conditioned on all other time points under the $m^{\text{th}}$ model.

$$g_t^{(m)} = \log p^{(m)}(\mathbf{y}_t | \mathbf{y}_{1,\cdots,t-1,t+1,\cdots,T}) \tag{36}$$

In other words, we attempt to initialize based on how well each Gaussian SSM predicts the current observation, $\mathbf{y}_t$, based on all the past and future observations. We expect this informative initialization of $g_t^{(m)}$ to be close to the global maximum in general. Since conventional Kalman filtering and RTS smoothing cannot compute the interpolated density easily, we utilize a different implementation (see S3 Appendix) [109].

This new initialization technique is well-grounded in the view of $g_t^{(m)}$ as the log-evidence of generating the observation at each time point within the HMM of a discrete-valued state $s_t = m$, with $m \in \{1, \cdots, M\}$. In the absence of known Gaussian SSM states, the next best choice is to evaluate which model dynamic provides the closest interpolation from all other time points for the current observation. It can also be seen as a "smoothing" extension of using filtered densities in place of the HMM observation probabilities in the early switching state-space model inference literature [34, 35, 75, 110].

As extensively analyzed in the Segmentation with posterior inference and Segmentation with parameter learning sections in Results, variational inference and learning with interpolated densities substantially improve over deterministic annealing and offer greater segmentation accuracy compared to the other switching inference methods. However, unlike the definition of $g_t^{(m)}$ in Eq 18 that is comparable across arbitrary Gaussian SSMs due to the identical $\mathbf{R}$, interpolated densities are bonafide normalized distribution density functions. Therefore, the model-specific parameters (e.g. $\mathbf{Q}^{(m)}$), especially the hidden state dimensionality, could bias the interpolated densities during the initialization. A simple heuristic is suggested in S3 Appendix to address this bias if present and shows robust performance in the spindle detection problem.

## Sleep spindle detection application

To demonstrate the utility of switching state-space models in neural signal processing, we analyzed EEG recorded during overnight sleep from a healthy young adult. The sleep EEG data were collected at the Massachusetts General Hospital Sleep Center after obtaining written

informed consent from the subject, and the study protocol was approved by the Massachusetts General Hospital Human Research Committee. The recording was acquired using the 128-channel eego mylab EEG amplifier system (ANT Neuro, Enschede, The Netherlands) at a sampling rate of 1 kHz. Ag/AgCl electrodes were arranged in an equidistant Waveguard montage (ANT Neuro). The ground and online reference electrodes were placed at the left mastoid and a central electrode Z3, respectively. Sleep staging was scored by a trained polysomnography technician following AASM guidelines [111].

EEG data segments were extracted from the periods scored as non-rapid eye movement (NREM) stage 2 sleep, re-referenced to the common average reference, and then downsampled to 100 Hz. We analyzed single-channel segments from a left parietal electrode LA2 (analogous to C3 in the International 10–20 system). EEG spectrograms were computed using the multitaper method [24] with 1 s window length and 95% overlap between adjacent windows (3 discrete prolate spheroidal sequences tapers, corresponding to a time-half-bandwidth product of 2, and $2^{10}$ minimum number of fast Fourier transform samples) after constant detrending within the sliding window.

**Initialization of Gaussian SSM parameters.** As mentioned in the Real-world application: Spindle detection section in Results, we modeled slow oscillations ($\delta$) and sleep spindles ($\varsigma$) observed in EEG recordings during NREM stage 2 sleep using Gaussian SSMs of the following form:

$$\begin{bmatrix} x_{t,1} \\ x_{t,2} \end{bmatrix} = a \begin{bmatrix} \cos\omega & -\sin\omega \\ \sin\omega & \cos\omega \end{bmatrix} \begin{bmatrix} x_{t-1,1} \\ x_{t-1,2} \end{bmatrix} + \begin{bmatrix} w_{t,1} \\ w_{t,2} \end{bmatrix}, \quad \begin{bmatrix} w_{t,1} \\ w_{t,2} \end{bmatrix} \sim \mathcal{N}\left(\mathbf{0}, \begin{bmatrix} \sigma^2 & 0 \\ 0 & \sigma^2 \end{bmatrix}\right) \quad (37)$$

The Gaussian SSM parameters, $\{a^\delta, \omega^\delta, (\sigma^2)^\delta\}$, $\{a^\varsigma, \omega^\varsigma, (\sigma^2)^\varsigma\}$, and the observation noise variance $R$ need to be initialized in order to learn the optimal model parameters using the generalized EM algorithm. We accomplished this by fitting two oscillators (one in slow frequency range, the other in spindle frequency range) to the EEG time series, assuming that both oscillations are present for the entire duration. This fitting was done using a standard EM algorithm [48, 93] with the parameters initialized based on our prior knowledge of the typical frequencies of these sleep oscillations:

$$a^\delta = 0.98 \qquad \omega^\delta = 2\pi\frac{1\,\text{Hz}}{100\,\text{Hz}} \qquad (\sigma^2)^\delta = 1$$

$$a^\varsigma = 0.98 \qquad \omega^\varsigma = 2\pi\frac{13\,\text{Hz}}{100\,\text{Hz}} \qquad (\sigma^2)^\varsigma = 1.$$

$$R = 1$$

Initial states were taken as zero-mean white noise with variance of 3 and not updated. We ran the EM algorithm for 50 iterations and used the resultant parameters as initial guesses for the Gaussian SSMs in switching state-space models. For the switching inference algorithms that do not have mechanisms to update model parameters, these parameters after the 50 EM iterations were directly used to infer segmentation.

**Priors for MAP estimates.** Parameters in state-space models of the form in Eq 37 can be subjected to prior distributions to yield MAP instead of ML estimation. We followed [48] to impose priors on the rotation frequency, $\omega$, and the state- and observation-noise variances, $\sigma^2$ and $R$ (see S3 Appendix). We used these MAP estimates throughout all the M-steps that involve updating the Gaussian SSM parameters and the observation noise variance.

## Supporting information

**S1 Appendix. Free energy calculation.**
(PDF)

**S2 Appendix. Joint estimation of parameters.**
(PDF)

**S3 Appendix. Implementation details.**
(PDF)

**S4 Appendix. Additional analyses.**
(PDF)

## Acknowledgments

We would like to thank Amanda M. Beck for useful discussions on modeling neural signals with oscillators.

## Author Contributions

**Conceptualization:** Mingjian He, Patrick L. Purdon.

**Data curation:** Mingjian He.

**Formal analysis:** Mingjian He, Proloy Das, Gladia Hotan, Patrick L. Purdon.

**Funding acquisition:** Patrick L. Purdon.

**Investigation:** Mingjian He, Proloy Das, Patrick L. Purdon.

**Methodology:** Mingjian He, Proloy Das, Gladia Hotan, Patrick L. Purdon.

**Project administration:** Mingjian He, Patrick L. Purdon.

**Resources:** Mingjian He, Proloy Das, Patrick L. Purdon.

**Software:** Mingjian He, Proloy Das, Gladia Hotan, Patrick L. Purdon.

**Supervision:** Proloy Das, Patrick L. Purdon.

**Validation:** Mingjian He, Proloy Das.

**Visualization:** Mingjian He, Proloy Das, Gladia Hotan.

**Writing – original draft:** Mingjian He, Proloy Das, Gladia Hotan, Patrick L. Purdon.

**Writing – review & editing:** Mingjian He, Proloy Das, Patrick L. Purdon.

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
