## [Decision Letter · Decision Letter 0]

10 Feb 2023

Dear Dr. Purdon,

Thank you very much for submitting your manuscript "Switching state-space modeling of neural signal dynamics" for consideration at PLOS Computational Biology.

As with all papers reviewed by the journal, your manuscript was reviewed by members of the editorial board and by several independent reviewers. In light of the reviews (below this email), we would like to invite the resubmission of a significantly-revised version that takes into account the reviewers' comments.

We cannot make any decision about publication until we have seen the revised manuscript and your response to the reviewers' comments. Your revised manuscript is also likely to be sent to reviewers for further evaluation.

Sincerely,

Xue-Xin Wei

Academic Editor

PLOS Computational Biology

Marieke van Vugt

Section Editor

PLOS Computational Biology

Reviewer's Responses to Questions

**Comments to the Authors:**

Reviewer #1: Manuscript by He and coworkers thoroughly describes a statistical method for time series segmentation. Switching linear state space modeling framework that combines linear-gaussian observation, linear-gaussian processes, and hidden-markov-model is a simple yet useful model for seemingly non-stationary processes. Although they have been known for decades, only recently, there has been a wider acceptance of such methods in neuroscience. The manuscript is well-written and has high pedagogical value as it meticulously describes the model, derivation, and numerical procedures. In my opinion, the main contributions of the work are in the methods and appendix sections. The main contribution is the new initialization scheme and mathematical elegance of the model that makes the switching linear state space model usable. Although the methods are evaluated for neural time series segmentation, the inference framework is general and can be used for estimating the continuous latent states as well. The code is made available online with the BSD 3-Clause Clear License.

Comments:

1. I enjoyed reading the manuscript very much.

1. I have an issue with the word ``non-stationary''. ``Time-varying activity'' is not a synonym with ``non-stationary'' process. In the strict mathematical definition of stationarity, all models and their marginals described in the paper are stationary, as the joint probability law are shift invariant in time. Stationary models can have transient events as long as the statistics of the transient events are time invariant.

1. A major weakness of the numerical results is the fixed observation matrix. The writing does not remark this fact. There are invariances that are taken care of with such assumption, and with sufficient domain knowledge it can be justified. However, this may not be the case in practice, and the inference framework presented does not assume a fixed observation matrix (eq. 28). I would like to see discussions regarding this.

1. Another major weakness of the numerical results is the lack of convergence studies. As a function of dataset size, how quickly do the parameters converge? For example, Fig 3. b, the distribution of R for VI-I inference seems to be almost the same as the initialization. Why is this the case? Does making the dataset longer resolve this? How much data is needed to have clearly superior segmentation results?

1. Fig 5. Only the joint VI-I is plotted. I would like to see the original parameterization as well.

1. The result section writing is not as smooth as the methods and appendices. Improvements in writing for presentation is suggested.

1. The EEG sleep spindle analysis is underwhelming. I would like to see the inferred parameters (with variational posterior distribution) and interpretations. Does the segmentation match expert evaluation? How predictive is the model?

1. The model used in spindle analysis is different from that in the simulation studies. This weakens the paper.

1. Fig 6. why are VI-I and VI-A posterior states so sharp in transition between 0 and 1? It looks like only those were thresholded at 0.5 and the other methods are reporting probability of spindle state directly.

1. p22. I disagree that you showed "extensive simulation results". The first two simulations were model matched validations. Only the third simulation provides some extra value. If the sample size analysis is added, perhaps it could be justified.

1. Discussion compares to relevant other methods. One thing that should be mentioned is the handling of the non-gaussian likelihoods. Many methods that are designed for neural spike train observations can handle Poisson process or other observation models.

1. p 36. "HMM with exp{g_t^(m)} as observation probabilities" <== normalization is necessary for such interpretation

1. p 41. M-step is closed-form, why say "convex optimization"?

1. S3 p4. It wasn't immediately clear to me that the interpolated densities corresponded to the equation. Perhaps adding an intermediate step could help.

1. The variational inference for each of the parameterized joint model seems to require some math. Please add a comment on how this may or may not be easily automated (perhaps using autodiff and black-box variational techniques)

Reviewer #2: Summary

In this paper, the authors advocate using switching state space models for modeling neural dynamics. They describe the specific switching state model under study and a variational inference algorithm used to fit the model. Methodologically, they then contribute a novel initialization method for the variational inference algorithm. The initialization procedure is validated and explored in simulated experiments, where they show improved recovery of ground truth segmentation and parameter learning compared to the previous initialization procedure. The authors then apply their model to sleep spindle detection.

The high-level approach of considering efficient and accurate inference algorithms for switching state space models of neural data is a worthwhile pursuit. Additionally, the proposed initialization procedure is interesting and promising. However, the paper could be improved in a number of areas described in detail below. Namely, the clarity of the contribution, situation of the work in the literature, and simulated experiments could be improved with some modifications.

Major comments and questions

1. It appears that the model and variational inference method are the same as those presented in Ghahramani and Hinton, 2000. However, at first I was under the impression that the proposed inference algorithm was developed by the authors in this paper. I suggest that the authors attempt to edit portions of the manuscript to make the relationship between the algorithm derived in this paper and the algorithm in Ghahramani and Hinton, 2000 more immediately clear. First, either 1) a statement saying the algorithm is the same as Ghahramani and Hinton or 2) these are the differences between the two algorithms (outside of initialization) would be helpful. Additionally, if it is correct that it is the same algorithm, the wording introducing the algorithm should be adjusted because it currently reads as if the authors developed this algorithm.

1a. For example, the abstract line “Here we derive a solution to the inference problem..” could be removed, and a new line describing the contribution as a new initialization procedure for a variational inference algorithm from [Ghahramani and Hinton, 2020] placed instead.

2. In simulation 2, the prior distributions used to sample initial parameter values could be improved. Currently, they are quite informative about the true parameter in that they are centered at the true parameter and typically come from non-overlapping distributions for the same parameter across states (e.g. F1 in U[0.8,1.0] and F2 in U[0.6,0.8]). I suspect this choice of distributions provides a lot of information that is unrealistic when the parameters are truly unknown. I suggest using distributions with wider support, not necessarily centered at the true value, and that the same parameter across states be drawn from the same distribution. These changes should help better assess how well each algorithm performs when the parameters are unknown.

3. More importantly, I think the modifications described above for simulation two will also be important in evaluating the initialization procedure. As opposed to deterministic annealing, it seems as if the initialization procedure relies more strongly on having good estimation of the parameters early in inference, since it is based on log probabilities under the current LDS models. So while the proposed initialization procedure has shown promising performance in simulation 1 with the true parameters known, and in simulation 2 where the initial parameter values are quite informative about the true parameters, it will be important to see how it performs in a more difficult setting.

4. In the section “extensions of model structure and parameter estimation” (line 184), it is worth noting that the authors are describing a switching linear dynamical system as described e.g. in ref 64, where the inference algorithm is designed to handle switches in the dynamics parameters. Therefore It would be helpful to compare an algorithm such as the one in ref 64. Additionally, I suggest changing initial parameter value distributions in this example as in the previous simulation.

5. The simulated examples and application all use models that are relatively low dimensionality in the state space. For example, it is often that there are 2 different LDS models each with 1 latent dimension. I would suggest that the authors explore how their initialization procedure performs as the number of LDS models and number of dimensions grows, compared to deterministic annealing, to help determine the overall significance of the initialization contribution.

Minor comments

- Could cite additional uses of switching state space models for neural data: Petreska et al., 2011; Zoltowski et al., 2020

- On pg. 33, I would recommend rephrasing “making exact inference possible”. It could mislead a reader into thinking the overall inference method is exact, whereas this really refers to algorithm subcomponents.

**Have the authors made all data and (if applicable) computational code underlying the findings in their manuscript fully available?**

Reviewer #1: Yes

Reviewer #2: Yes

PLOS authors have the option to publish the peer review history of their article (what does this mean?). If published, this will include your full peer review and any attached files.

Reviewer #1: No

Reviewer #2: No
---

## [Decision Letter · Decision Letter 1]

23 May 2023

Dear Dr. Purdon,

Thank you very much for submitting your manuscript "Switching state-space modeling of neural signal dynamics" for consideration at PLOS Computational Biology.

As with all papers reviewed by the journal, your manuscript was reviewed by members of the editorial board and by several independent reviewers. In light of the reviews (below this email), we would like to invite the resubmission of a significantly-revised version that takes into account the reviewers' comments.

We cannot make any decision about publication until we have seen the revised manuscript and your response to the reviewers' comments. Your revised manuscript is also likely to be sent to reviewers for further evaluation.

Sincerely,

Xue-Xin Wei

Academic Editor

PLOS Computational Biology

Marieke van Vugt

Section Editor

PLOS Computational Biology

Reviewer's Responses to Questions

**Comments to the Authors:**

Reviewer #1: Thank you for the updated manuscript. It addresses most of my concerns adequately. The additional analyses are much appreciated. I have just one remaining concern in response to the previous point no 9.

## Posterior discrete state concentration during the fixed-point iteration

It is intriguing that the approximate, factorized posterior tends to converge to a very sparse state through the fixed-point iteration inner loop. The form of the reciprocals is akin to the sparse Bayesian Gaussian regression and factor analysis procedures know as ARD. Automatic Relevance Determination (ARD) which is an iterative type-II maximum likelihood procedure leads to sparsification of coefficient in a similar iterative manner. If there's a relationship that can be shown, I suggest adding some discussion. ARD has been popularized by Bishop's PRML book and has been used in neuroscience widely.

Although it is nice to have a sparse selection of discrete states after the inference, this is clearly not always going to be close to the true posterior. I don't think this is necessarily a bad thing, but it needs to be made clear that the space of potential approximate posteriors are smaller than postulated by the explicit assumptions.

The new methods hidden in S4, "Soft and hard segmentation with VI-I EM", can be useful but they do not represent principled derivations. How would one quantify the usefulness of the soft segmentation method? I would like to see some discussion for the practitioner. Also, please make sure the code includes these soft/hard segmentation options for ease of use.

Minor:

## There are errors in the citation, e.g.,

Glaser JI, Whiteway M, Cunningham JP, Paninski L, Linderman SW. Recurrent

Switching Dynamical Systems Models for Multiple Interacting Neural

Populations. Neuroscience; 2020.

 published in "Advances in Neural Information Processing Systems"

Reviewer #2: Thank you for your thorough response and revisions. I still have some lingering, major concerns though.

# Concern 1: Bringing interpolated density and parameter initialization to the fore

a. The revised manuscript makes clear that the methodological novelty is in the initialization scheme, but the main text gives very few details of the interpolated density initialization scheme. Instead, the details are punted to Supplement S3. Given that this is a core contribution, it should be a major component of the main text. The relative fraction of main text devoted to describing generalized EM as opposed to describing initialization strategies is not commensurate with the relative novelty of these two facets of the work.

b. A skeptical reader could ask whether the generalized EM algorithm is necessary at all, with reasonable initial estimates of the parameters. Given the same initial parameters $\\theta^{(0)}$, one could used blocked Gibbs sampling (and many previous works have used blocked Gibbs to great success on SLDS, as the authors note). The analog of interpolated density initialization would seem be: initialize the continuous states with samples from the conditional distribution $p(x_{1:T}^{(m)} | y_{1:T}, s_t=m \\forall t, \\theta^{(0)})$, and then sample $s_{1:T}$ from its conditional distribution. With weak, conjugate priors on the parameters and good initialization, I would expect blocked Gibbs sampling to work very well. I realize that doing this comparison is non-trivial so I don't consider it a must-have, but I would personally be interested to know the outcome. (FWIW, https://github.com/mattjj/pyslds has Gibbs sampling with conjugate priors implemented for SLDS, so some of the annoying parts of working with priors on matrix-valued parameters could be ameliorated.)

# Concern 2: Framing of existing methods

The discussion presents existing works as "black-box variational inference" methods when in fact many of the cited references (39, 41, 71, 76) use structure exploiting inference algorithms like blocked Gibbs sampling (39, 41) or structured mean field approximations (71, 76). Both sets of algorithms exploit conditional conjugacy to update blocks of random variables at once, and they do so with exact updates rather than stochastic gradient ascent steps. The structured mean field approaches are directly analogous to the structured mean field approach used in this paper. The main difference is not black box vs structured, but rather the initialization scheme and the model/posterior family itself. T models and posterior approximation in this paper have a factorial construction with many continuous states evolving in parallel, rather than one continuous state with switching dynamics, as in the cited works.

# Concern 3: Switching SSM Framing in the Introduction

I disagree with the paragraph on lines 47-59 of the introduction.

- "These methods apply carefully constrained approximations to represent a broader range of switching processes besides the Markov process used in the HMM" My impression is that the cited work uses the same sorts of approximations as are developed in this submission, and they target very similar models (switching linear dynamical systems) with some extensions (nonparametric models, continuous->discrete dependencies, etc.)

- "Markov chain Monte Carlo sampling techniques are used to perform inference and learning for these models, which may be computationally burdensome for large data sets collected in neuroscience studies." The complexity of blocked Gibbs updates is identical to those of generalized EM. In practice, I've found blocked Gibbs often yields comparable estimates to structured mean field in the same wall-clock time, and can even be more robust to local optima than generalized EM.

- "Moreover, the nonparametric formulation of these algorithms may not be easily interpretable in relation to the neural mechanisms being studied." This seems quite speculative.

- "Similarly, under these methods it is difficult to constrain the switching inferences to candidate models formed based on clinical or neuroscience domain knowledge." I don't see why the cited models couldn't be constrained in the same way as done here.

- "Perhaps due to these challenges, these more recent Bayesian algorithms have also not been widely adopted to study neural data with time-varying dynamics." There has been a lot of recent work using switching linear dynamical systems to neural data with time-varying dynamics. In addition to the works cited already in the paper, see also:

- Taghia et al (Nature Comm, 2019) https://www.nature.com/articles/s41467-018-04723-6

- Nair et al (Cell, 2023) https://www.cell.com/cell/pdf/S0092-8674(22)01471-4.pdf

- Large body of work from Emily Fox on Autoregressive HMMs for EEG analysis and Sandeep Datta's lab using AR-HMMs for behavioral modeling. These models are not the same as SLDS, but they are closely related and demonstrate the breadth with which switching state space models are used in neuroscience.

I realize this is just one paragraph, but I think it unfortunately misconstrues the existing work in this domain.

# Concern 4: Balance of simulated vs real data examples

In my opinion, the most exciting part of this submission is not the algorithm derivation or the initialization scheme, but rather the application to spindle detection. I think it's very exciting to see these methods performing well on a non-trivial and clearly important scientific problem. Besides rebalancing the methodological section (Concern 1), I would personally prefer to see more emphasis placed on the applications and less on the relative comparison of initialization schemes.

**Have the authors made all data and (if applicable) computational code underlying the findings in their manuscript fully available?**

Reviewer #1: **No: **Initial submission code is available, but updated revision related codes are not in the same repo. E.g. the soft/hard segmentation code doesn't seem to be shared.

Reviewer #2: None

PLOS authors have the option to publish the peer review history of their article (what does this mean?). If published, this will include your full peer review and any attached files.

Reviewer #1: No

Reviewer #2: No
---

## [Decision Letter · Decision Letter 2]

28 Jul 2023

Dear Dr. Purdon,

We are pleased to inform you that your manuscript 'Switching state-space modeling of neural signal dynamics' has been provisionally accepted for publication in PLOS Computational Biology.

Best regards,

Xue-Xin Wei

Academic Editor

PLOS Computational Biology

Marieke van Vugt

Section Editor

PLOS Computational Biology

Reviewer's Responses to Questions

**Comments to the Authors:**

Reviewer #1: Thanks for your thoughtful responses. I am satisfied.

Reviewer #2: Thank you for your thorough responses. Your new changes address all of my concerns. I think this work is a nice addition to the literature on switching state space models for neural data analysis. Congratulations on a very nice paper!

**Have the authors made all data and (if applicable) computational code underlying the findings in their manuscript fully available?**

Reviewer #1: Yes

Reviewer #2: None

PLOS authors have the option to publish the peer review history of their article (what does this mean?). If published, this will include your full peer review and any attached files.

Reviewer #1: No

Reviewer #2: No

---

## [Editor Report · Acceptance letter]

18 Aug 2023

PCOMPBIOL-D-22-01692R2 

Switching state-space modeling of neural signal dynamics

Dear Dr Purdon,

I am pleased to inform you that your manuscript has been formally accepted for publication in PLOS Computational Biology. Your manuscript is now with our production department and you will be notified of the publication date in due course.

With kind regards,

Lilla Horvath
